# PACR: PROGRESSIVELY ASCENDING CONFIDENCE REWARD FOR LLM REASONING

## ABSTRACT

Reinforcement Learning with Verifiable Rewards (RLVR) has significantly improved LLM reasoning, but its sparse, outcome-based reward provides no guidance for intermediate steps, slowing exploration. We propose **P**rogressively **A**scending **C**onfidence **R**eward (**PACR**), a dense, model-intrinsic reward computed directly from the model's evolving belief in the correct answer. PACR encodes the inductive bias that, along a well-formed reasoning trajectory, the probability of the ground-truth answer should have a generally ascending trend. We provide empirical and theoretical analysis validating that such an inductive bias constrains the exploration search space to regions richer in logically sound reasoning. We demonstrate that PACR accelerates exploration, reaches reward saturation with fewer trajectories, and yields improvements on multiple benchmarks. Our results suggest that dense, model-intrinsic shaping signals can make RLVR training more effective and reliable. Code will be released.

## 1 INTRODUCTION

Pre-trained large language models (LLMs) exhibit strong performance on complex, multi-step reasoning tasks (Comanici et al., 2025; Yang et al., 2025a; Team, 2025). Reinforcement Learning with Verifiable Rewards (RLVR) has emerged as a leading approach for further improving such capabilities, using a programmatically checkable terminal metric (e.g., exact-match on the final answer) as the reward (Shao et al., 2024b; Guo et al., 2025). While effective, the standard RLVR formulation supplies a sparse terminal accuracy signal, offering no guidance for intermediate steps and thus exacerbating credit assignment. Alternative process-based supervision employs external reward mod-

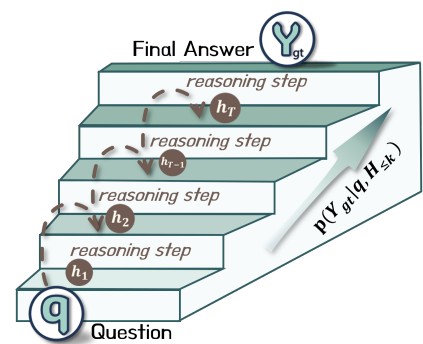

Figure 1: **Stepwise confidence growth.** For a question $q$, a well-formed sequence of reasoning steps $h_1, \ldots, h_k$ should increase the model's probability of the ground-truth answer $Y_{\text{gt}}$ across steps.

els to score intermediate reasoning, but is costly to train, data-hungry, and prone to misalignment (Cui et al., 2025; Cheng et al., 2025).

This work asks whether we can obtain *stepwise supervision* directly from the model. Psycholinguistic work shows that people interpret language incrementally, updating expectations with each word; as context accumulates, uncertainty falls and the correct interpretation becomes more likely (Hale, 2001; Levy, 2008). By the same logic, in tasks with a verifiable final answer, a correct intermediate step should typically raise the model's probability of the ground-truth answer. Concretely, given a question $q$, a reasoning prefix $H_{\leq k}$, and ground truth $Y_{\text{gt}}$, we track the model's confidence $p(Y_{\text{gt}} \mid q, H_{\leq k})$ and expect a general upward trend over steps (Figure 1).

Guided by this premise, we introduce the **Progressively Ascending Confidence Reward (PACR)**, a dense, model-intrinsic signal that converts confidence growth into stepwise supervision for LLM reasoning during reinforcement learning. During training, as the model produces a sequence of reasoning steps for a question with a verifiable answer, we evaluate at each step the log-probability assigned to the ground-truth answer and reward any positive change, effectively encouraging a consis-

tently upward trend in confidence. Because PACR is computed from the model's own probabilities, it requires no external reward model and is available at every step, improving credit assignment and steering search toward faithful trajectories. We pair PACR with the standard RLVR terminal accuracy reward so the objective remains anchored to verifiable correctness while the process signal shapes the reasoning path. **In detail, our contributions can be summarized as follows:**

- **Empirical Validation of an Inductive Bias (Section 4.1).** We provide extensive observational evidence that ground-truth confidence growth acts as a powerful inductive bias. Our analyses on open-source LLMs reveal three key findings: (1) a *consistent* confidence ascent strongly correlates with final answer correctness; (2) among correct answers, logically coherent reasoning paths exhibit an even *more consistent* ascent than spurious ones; and (3) the *magnitude* of the confidence gain effectively pinpoints pivotal reasoning steps.
- **Theoretical Justification (Section E and 5).** We provide a theoretical foundation for using confidence growth as a process reward. We prove that a reasoning step from an idealized oracle policy will, on average, increase or maintain the model's confidence in the ground truth, validating it as a strong inductive bias. Building on this, we formalize the **Progressively Ascending Confidence Reward (PACR)** and introduce two concrete methods for its implementation: **Sparse-PACR** for trajectory-level rewards and **Dense-PACR** for step-wise rewards.
- **Experimental Results (Section 7).** Across multiple reasoning benchmarks, augmenting RLVR with our PACR methods improves training dynamics and final performance. Our approach accelerates exploration and ultimately attains a higher, more consistent final score than the baseline, demonstrating a more effective and reliable training process.

## 2 RELATED WORK

**Outcome-based RL for LLM Reasoning** Reinforcement Learning (RL) is increasingly used to fine-tune Large Language Models (LLMs). This is done not only to align models with human preferences via Reinforcement Learning from Human Feedback (RLHF) (Ouyang et al., 2022; Li et al., 2024b; Bai et al., 2022) but also to enhance their reasoning abilities for complex problem-solving (Kumar et al.). To improve these reasoning capabilities, a recent prominent approach is Reinforcement Learning with Verifiable Reward (RLVR) (Guo et al., 2025; Yang et al., 2024; Shao et al., 2024a), which uses an outcome-based reward instead of a proxy reward model. For example, a reward of 1 is assigned for a correct answer and 0 (or -1) for an incorrect one. Then, the model generates multiple trajectories for a single problem. The reward for each trajectory is then compared against the average reward across all samples in the group, and this relative reward is used as an advantage to train the model. This outcome-based reward system is widely explored (Liu et al., 2025b; Yu et al., 2025; Hu et al., 2025; Zeng et al., 2025b) because it is easily scalable, and mitigates concerns about reward hacking by eliminating the need for a separate reward model (Guo et al., 2025). However, this approach has a significant limitation for complex reasoning tasks that require generating a long thought process (Zhang et al., 2025). In such cases, relying solely on the final outcome provides a sparse and noisy reward signal.

**Dense Reward for LLMs Finetuning with RL** To overcome the limitations of holistic, trajectory-level sparse rewards, various approaches for providing dense rewards have been explored. In RLHF, for instance, approaches include training an external reward model to assign token-level rewards using synthesized data (Yoon et al., 2024), utilizing a more mature external model as the reward model (Cao et al., 2024; Wu et al., 2023), and use implicit reward signal from reward model (Chan et al., 2024). Similarly, direct alignment algorithms (e.g., DPO (Rafailov et al., 2023)) have been adapted to provide dense rewards by re-framing DPO's implicit reward at a token level (Zeng et al.; Zhu et al.; Zhong et al.; Rafailov et al.) or by selectively using specific tokens for the reward signal (Yoon et al.; Liu et al.). For training a reasoning LLM with RL, previous approaches include training a Process Reward Model (PRM) for process-level rewards (Li & Li, 2025; Cheng et al., 2025; Zhang et al., 2025), or defining a DPO-like implicit reward at the token level (Cui et al., 2025; Yuan et al., 2024). However, these approaches typically require additional models, such as a reward model or a reference model, to generate the reward signal. In contrast, our work eliminates the need for any additional models. We instead use the current policy model itself to generate a dense reward signal that enhances reasoning capabilities.

## 3 BACKGROUND AND PROBLEM SETUP

This section introduces the notation for reasoning trajectories, how we segment and evaluate stepwise confidence in the ground-truth answer, and the RL objective we use in training.

**Reasoning Trajectories and Notation.** Given a question $q$, a policy $\pi_\theta$ generates a *sequence of reasoning steps* $H = (h_1, \ldots, h_T)$ and a final answer $\hat{Y}$. Let $Y_{\text{gt}}$ denote the verifiable ground-truth answer. We write $H_{\leq k} = (h_1, \ldots, h_k)$ for the reasoning steps up to step $k$. We analyze and shape the reasoning process by tracking how the model's probability of $Y_{\text{gt}}$ evolves with the prefix $H_{\leq k}$.

**Segmenting the Reasoning Process and Stepwise Ground-truth Confidence.** Similar to Yang et al. (2025c), we segment each generated reasoning trace into discrete steps $\{h_k\}_{k=1}^T$ using a simple, model-agnostic rule: start a new step at a newline ('\n') or at a period followed by a space ('. '); fragments shorter than five tokens are merged with the preceding step to avoid overly fine splits (Further discussion on segmentation strategies is provided in Appendix B). To measure ground-truth–anchored confidence at step $k$, we standardize the answer format by appending a short prefix $y_{\text{gt}}^0$ (e.g., 'So the final answer is \boxed{') and evaluate the model's probability of the ground-truth answer $Y_{\text{gt}} = (y_{\text{gt}}^1, \ldots, y_{\text{gt}}^L)$ under the current prefix $H_{\leq k}$. Writing $Y_{\text{gt}} = (y_{\text{gt}}^1, \ldots, y_{\text{gt}}^L)$, we measure the **ground-truth confidence** at step $k$ as

$$\log p(Y_{gt}|q, H_{\leq k}) \ = \ \sum_{l=1}^{L} \log p_\theta\big(y_{\text{gt}}^l \mid q,\, H_{\leq k},\, y_{\text{gt}}^0,\, y_{\text{gt}}^{<l}\big), \tag{1}$$

where $y_{\text{gt}}^{<l}$ are preceding answer tokens. This measures the model's confidence in the ground truth answer at any given stage of its reasoning steps.

**Group Relative Policy Optimization (GRPO)** GRPO (Shao et al., 2024b) estimates advantages by comparing returns *within* a group of $N$ samples rather than using a learned value function. For a given question $q$ (with verifiable answer $Y_{\text{gt}}$), the behavior policy $\pi_{\theta_{\text{old}}}$ generates $N$ trajectories

$$\{\tau^{(i)}\}_{i=1}^N, \qquad \tau^{(i)} \ = \ \big(H^{(i)}, \hat{Y}^{(i)}\big), \tag{2}$$

where $H^{(i)} = (h_1^{(i)}, \ldots, h_{T_i}^{(i)})$ are the reasoning steps, $T_i$ is the number of steps for $i$-th trajectory and $\hat{Y}^{(i)}$ is the predicted answer for $i$-th trajectory.

For each sampled trajectory $i$, we compare the predicted answer $\hat{Y}^{(i)}$ with the ground truth $Y_{\text{gt}}$ and assign a binary terminal accuracy reward:

$$R^{(i)} \ = \ \begin{cases} 1, & \texttt{is\_equivalent}\big(\hat{Y}^{(i)}, Y_{\text{gt}}\big) \\ 0, & \text{otherwise.} \end{cases} \tag{3}$$

Here `is_equivalent` performs task-specific normalization (e.g., stripping whitespace/punctuation, handling LaTeX boxing, case folding, and numeric tolerances) before exact match. The group-relative advantage for trajectory $i$ is computed by centering (and optionally standardizing) its reward within the cohort of $N$ samples:

$$A^{(i)} = \frac{R^{(i)} - \text{mean}(\{R^{(i)}\}_{i=1}^N)}{\text{std}(\{R^{(i)}\}_{i=1}^N)}. \tag{4}$$

Similar to PPO (Schulman et al., 2017), GRPO adopts a clipping with KL penalty:

$$\mathcal{J}_{\text{GRPO}}(\theta) =$$

$$\mathbb{E}_{\substack{(q, Y_{gt}) \sim \mathcal{D} \\ \{\tau^{(i)}\} \sim \pi_{\theta_{\text{old}}}(\cdot|q)}} \left[ \frac{1}{N} \sum_{i=1}^N \frac{1}{|\tau^{(i)}|} \left( \min\Big( \frac{\pi_\theta(\tau^{(i)} \mid q)}{\pi_{\theta_{\text{old}}}(\tau^{(i)} \mid q)}(\theta) A^{(i)},\, \text{clip}\Big(\frac{\pi_\theta(\tau^{(i)} \mid q)}{\pi_{\theta_{\text{old}}}(\tau^{(i)} \mid q)}, 1-\varepsilon, 1+\varepsilon\Big) A_i \Big) - \beta D_{\text{KL}}(\pi_\theta || \pi_{\text{ref}}) \right) \right], \tag{5}$$

where $\mathcal{D}$ is the training dataset and $\pi_{\text{ref}}$ is a reference policy. In our work, we follow the Dr. GRPO (Liu et al., 2025b) formulation, a bias-mitigated variant of GRPO. This approach modifies the standard GRPO algorithm by discarding the standard deviation from the advantage calculation and the length normalization from the loss function (the terms shown in **green** in Eq. 4 and Eq. 5).

## 4  IS GROUND-TRUTH CONFIDENCE GROWTH A USEFUL INDUCTIVE BIAS?

We posit that reasoning fundamentally functions as a process of *uncertainty reduction*. A faithful reasoning step provides intermediate evidence that bridges the gap to the solution, mathematically manifesting as an increase in the probability of the ground-truth answer. This suggests the following inductive bias for learning: valid reasoning steps should be characterized by **positive confidence gain on the ground-truth answer**.

To validate this inductive bias as a reward, we test two key conditions:

a) **Granular Quality:** Does the ground-truth confidence gain correlate with step-level reasoning quality?

b) **Causal Utility:** Does guiding generation with this bias improve accuracy during inference?

**Ground-truth Confidence Growth.**   We first quantify confidence growth by defining the stepwise confidence gain, $C_k$, as the change in the log-probability of the ground-truth answer induced by the addition of reasoning step $h_k$:

$$C_k := \log \pi_\theta\big(Y_{\text{gt}} \mid q, H_{\leq k}\big) \; - \; \log \pi_\theta\big(Y_{\text{gt}} \mid q, H_{<k}\big), \tag{6}$$

where $H_{\leq k} = (h_1, \ldots, h_k)$ and $H_{<k} = (h_1, \ldots, h_{k-1})$. For $k = 1$, $H_{<1}$ is the empty prefix. Intuitively, $C_k$ measures the information gain regarding the ground truth provided by step $h_k$. (When indexing trajectories, we write $C_k^{(i)}$.) For brevity, we will hereafter use "*confidence growth*" and "*ground-truth confidence growth*" interchangeably.

### 4.1  OBSERVING GROUND-TRUTH CONFIDENCE GROWTH ON REASONING MODELS

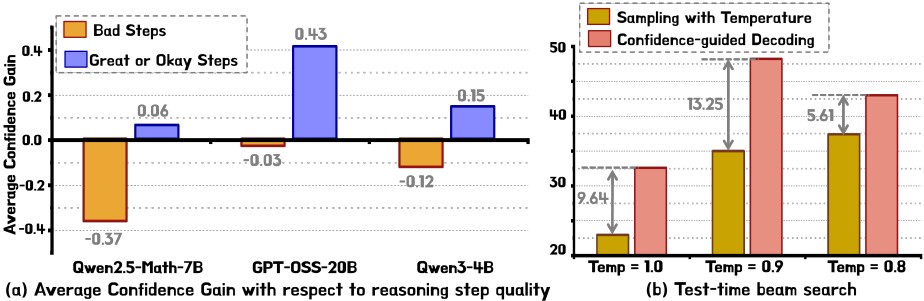

Figure 2: **Validation of Confidence Growth Utility. (a) Granular Quality:** We utilized GPT-5.1 to annotate individual reasoning steps as GREAT, OKAY, or BAD. We then analyzed the model's **intrinsic** confidence gain ($C_k$) for each category. The results show that high-quality steps consistently drive positive $C_k$, while flawed steps yield negligible or negative gains. **(b) Causal Utility:** Using $C_k$ to guide generation (via beam search with width 1) on Qwen2.5-Math-7B significantly improves accuracy compared to standard sampling across multiple temperatures. This confirms that maximizing confidence growth actively steers the model toward correct solutions.

**a) Granular Quality: Does the ground-truth confidence gain correlate with step-level reasoning quality?**   To demonstrate that $C_k$ effectively captures reasoning quality of a step, we performed a fine-grained analysis using trajectories sampled from diverse models. We utilized an external verifier to annotate individual reasoning steps, classifying each into one of three categories based on the definitions established by Lightman et al. (2023) (refer to Appendix F for full annotation details):

- GREAT: A strong, logically sound step that makes meaningful mathematical progress.
- OKAY: A valid but low-value step (e.g., restating information or stalling) that adds minimal insight.
- BAD: A logically flawed, incoherent, or hallucinated step that leads the solution astray.

We analyzed the distribution of confidence gains ($C_k$) for each category. As illustrated in Figure 2-(a), we observe a strict trend:

Question: Compute $a + b + c$, given that $a$, $b$, and $c$ are the roots of $\frac{1}{x} + 5x^2 = 6x - 24$.

Reasoning:

$h_1$ | $C_1: -1.41$ | We need to solve equation: $\frac{1}{x} + 5x^2 = 6x - 24$.

$h_2$ | $C_2: +0.37$ | Multiply by $x$: $1 + 5x^3 = 6x^2 - 24x$.

$h_3$ | $C_3: -0.87$ | Bring all terms: $5x^3 - 6x^2 + 24x + 1 = 0$.

$h_4$ | $C_4: 0.00$ | So cubic polynomial: $5x^3 - 6x^2 + 24x + 1 = 0$.

$h_5$ | $C_5: +1.28$ | Roots $a, b, c$.

$h_6$ | $C_6: +0.78$ | We need sum $a + b + c$.

$h_7$ | $C_7: +4.83$ | For cubic $ax^3 + bx^2 + cx + d = 0$, sum of roots=$-b/a$.

$h_8$ | $C_8: -0.23$ | Here $a = 5, b = 6$.

$h_9$ | $C_9: +0.63$ | So sum $= -(-6)/5 = 6/5$? Wait polynomial: $5x^3 - 6x^2 + 24x - 1 = 0$.

$h_{10}$ | $C_{10}: -0.06$ | So coefficient of $x^2$ is $-6$.

$h_{11}$ | $C_{11}: +0.12$ | Sum of roots $= -(-6)/5 = 6/5$.

$h_{12}$ | $C_{12}: -0.05$ | So answer $6/5$.

Figure 3: **Qualitative example of a pivotal step.** Among the reasoning steps, a critical insight at step $h_7$ (the introduction of Vieta's formulas for a cubic equation) results in a large, distinct spike in the ground-truth confidence gain ($C_7 = +4.83$). This is substantially larger than the gains from more routine algebraic steps. Further qualitative examples are provided in Appendix E.

$$\text{Avg } C_k(\textsc{Great or Okay}) > \text{Avg } C_k(\textsc{Bad})$$

The analysis shows that GREAT or OKAY steps, on average, drive ground-truth confidence upward, whereas BAD steps fail to contribute valid evidence, resulting in negligible or negative gains. This validates $C_k$ as a dense, high-resolution signal capable of penalizing local errors and rewarding critical insights, a distinction that standard sparse outcome-based rewards fail to capture.

**b) Causal Utility: Does guiding generation with this bias improve accuracy during inference?**
To establish that confidence growth actively *guides* the reasoning process toward correctness, we utilized the stepwise confidence gain ($C_k$) as a scoring function for **test-time search**.

We implemented a **beam search with a beam width of 1**. At each reasoning step, we sampled $N = 8$ candidate extensions and greedily selected the single path maximizing the confidence gain $C_k$ to continue generation. **We note that since $C_k$ relies on the ground truth, this experiment serves purely as an analytical validation of the reward signal, not as a proposed inference method.**

We evaluated this approach across multiple sampling temperatures ($T \in \{0.8, 0.9, 1.0\}$). As shown in Figure 2-(b), this confidence-guided search consistently outperforms the standard sampling baseline across all temperature settings. These results indicate that $C_k$ serves as a robust steering signal, providing the dense supervision needed to differentiate valid paths from incorrect ones regardless of generation stochasticity. Crucially, this validates the confidence growth as a reinforcement learning reward: since the signal successfully guides the model to the correct solution when available, optimizing it during training encourages the model to intrinsically internalize this reasoning behavior.

**Large Stepwise Confidence Gains Pinpoint Pivotal Reasoning Steps.** Beyond the overall trend of confidence, we investigated whether the *magnitude* of the stepwise gain, $C_k$, correlates with the importance of individual reasoning steps. Qualitatively, we observe that large, positive spikes in $C_k$ often coincide with pivotal moments in the reasoning process, such as the application of a key theorem or a critical insight. For instance, as illustrated in Figure 3, a step introducing the sum of roots formula for a cubic equation yields a substantially larger confidence gain compared to adjacent steps involving routine algebraic manipulation. To validate this rigorously, we conducted a quantitative pairwise comparison. For trajectories in $\mathcal{T}_{\text{correct}}$, we randomly sampled pairs of reasoning steps, $h_i$

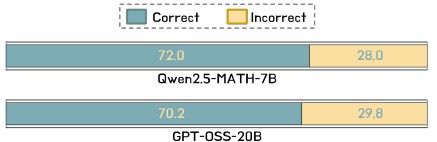

Figure 4: **Quantitative validation of step importance.** In a pairwise comparison, an LLM evaluator judged the step with the higher confidence gain ($C_i > C_j$) as more critical with a win rate significantly above chance, confirming that gain magnitude correlates with step importance.

and $h_j$, under the condition that $C_i > C_j$. We then prompted an LLM evaluator (GPT-5) to judge which of the two steps was more critical for reaching the final solution (see Appendix D for detailed evaluation prompts). The step with the higher confidence gain, $h_i$, was frequently identified as more critical, achieving a win rate significantly above chance (Figure 4). This finding suggests that the magnitude of the confidence gain is not arbitrary; it is a meaningful signal that effectively pinpoints influential steps within a reasoning chain.

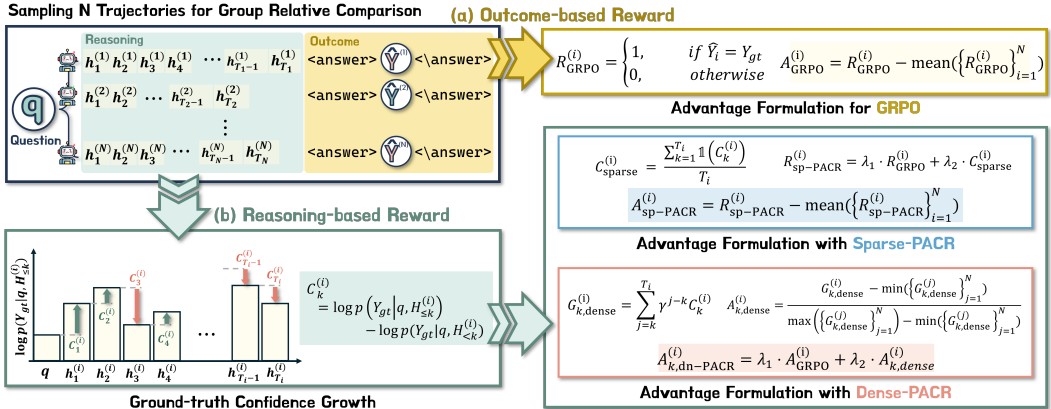

Figure 5: **Overview of the PACR method and its integration with GRPO.** Standard GRPO begins by sampling a group of $N$ reasoning trajectories for a given question. (a) A standard outcome-based reward ($R_{\text{GRPO}}^{(i)}$) is calculated based on the correctness of the final answer. (b) Our proposed reasoning-based reward is derived from the ground-truth confidence growth ($C_k^{(i)}$) at each step. This signal is integrated into the final advantage calculation in two ways: **Sparse-PACR** uses the consistency of confidence growth to compute a single reward for the entire trajectory, while **Dense-PACR** uses the magnitude of each step's gain to compute a fine-grained, per-step advantage.

## 5 METHOD: PROGRESSIVELY ASCENDING CONFIDENCE REWARD (PACR)

Based on our findings in Section 4, we now formalize how to incorporate the principle of ascending ground-truth confidence into the GRPO framework. To do this, we introduce the Progressively Ascending Confidence Reward (PACR), a procedural reward signal designed to complement the final outcome-based reward. We propose *two* variants: (1) **Sparse-PACR**, which applies a holistic, trajectory-level reward based on the consistency of confidence growth, and (2) **Dense-PACR**, which provides a fine-grained, step-wise reward based on the magnitude of each confidence change.

**Sparse-PACR.** In the Sparse setting, we compute a single procedural reward for an entire trajectory based on the consistency of its confidence growth. This reward, $C_{\text{sparse}}^{(i)}$, is the proportion of reasoning steps that produce a positive confidence gain. We calculate it using an indicator function, $\mathbb{I}(\cdot)$:

$$C_{\text{sparse}}^{(i)} = \frac{1}{T_i} \sum_{k=1}^{T_i} \mathbb{I}\Big(C_k^{(i)} > 0\Big), \tag{7}$$

where $C_k^{(i)}$ is the confidence gain in Eq. 6. The final reward for trajectory $i$, $R_{\text{sp-PACR}}^{(i)}$, is a weighted combination of the standard outcome-based reward, $R_{\text{GRPO}}^{(i)}$, and our sparse procedural reward:

$$R_{\text{sp-PACR}}^{(i)} = \lambda_1 \cdot R_{\text{GRPO}}^{(i)} + \lambda_2 \cdot C_{\text{sparse}}^{(i)}. \tag{8}$$

This combined reward is then used to calculate the trajectory's advantage, $A_{\text{sp-PACR}}^{(i)}$, within the GRPO framework by centering it against the group average:

$$A_{\text{sp-PACR}}^{(i)} = R_{\text{sp-PACR}}^{(i)} - \text{mean}\big(\{R_{\text{sp-PACR}}^{(j)}\}_{j=1}^N\big). \tag{9}$$

**Dense-PACR.** The Dense setting provides a more granular, step-wise reward signal. At each reasoning step $k$ in trajectory $i$, we use the ground-truth confidence gain, $C_k^{(i)}$, as an immediate reward. From this, we compute the discounted return for that step, $G_{k,\text{dense}}^{(i)}$, by summing the rewards from that point forward:

$$G_{k,\text{dense}}^{(i)} = \sum_{j=k}^{T_i} \gamma^{j-k} C_j^{(i)}, \tag{10}$$

where $\gamma$ is a discount factor. To create a stable, step-wise advantage signal, $A_{k,\text{dense}}^{(i)}$, we normalize these returns across the group at each step $k$. Specifically, we use Min-Max scaling to map the returns to a $[0, 1]$ range. This creates a purely positive signal that only incentivizes confidence growth without penalizing steps that do not, a design choice we validate in our ablations (Section 7.5). To handle trajectories of varying lengths, the discounted return $G_{k,\text{dense}}^{(i)}$ is treated as zero for any step $k$ that does not exist in trajectory $i$. The resulting advantage for a step $k$ in trajectory $i$ is then:

$$A_{k,\text{dense}}^{(i)} = \frac{G_{k,\text{dense}}^{(i)} - \min_j(\{G_{k,\text{dense}}^{(j)}\}_{j=1}^N)}{\max_j(\{G_{k,\text{dense}}^{(j)}\}_{j=1}^N) - \min_j(\{G_{k,\text{dense}}^{(j)}\}_{j=1}^N)}. \tag{11}$$

Finally, the total advantage at each step, $A_{k,\text{dn-PACR}}^{(i)}$, is the weighted sum of the trajectory-level GRPO advantage and our dense, step-wise advantage:

$$A_{k,\text{dn-PACR}}^{(i)} = \lambda_1 \cdot A_{\text{GRPO}}^{(i)} + \lambda_2 \cdot A_{k,\text{dense}}^{(i)}, \tag{12}$$

where $A_{\text{GRPO}}^{(i)} = R_{\text{GRPO}}^{(i)} - \text{mean}(\{R_{\text{GRPO}}^{(j)}\}_{j=1}^N)$. This final advantage is then used to update the policy.

# 6 EXPERIMENTAL SETUP

**Models and Baselines.** We experiment with three open-source LLMs: Qwen2.5-Math-1.5B, Qwen2.5-Math-7B (Yang et al., 2024), and Qwen3-4B[1] (Yang et al., 2025a). Our baseline for all experiments is Dr.GRPO (Liu et al., 2025b), a bias-mitigated version of GRPO (Shao et al., 2024b), which we implement using the OAT framework (Liu et al., 2024). We compare this baseline against our two proposed methods, Sparse-PACR and Dense-PACR.

**Datasets and Evaluation.** For training, we use the MATH dataset (Hendrycks et al.). Following prior work (Liu et al., 2025b), we use the full dataset for the 1.5B model and filter for the more challenging levels (3-5) for the 4B and 7B models. To evaluate performance, we test our models on five diverse mathematical reasoning benchmarks: MATH500 (Hendrycks et al.), Minerva-Math (Lewkowycz et al., 2022), OlympiadBench (He et al., 2024), AIME 2024, and AMC 2023 (Li et al., 2024a). Final answers are programmatically checked for correctness using the Math-Verify (Kydlíček, 2025) library. All results are reported as pass@1 using greedy decoding (temperature of 0).

**Training Details.** For each problem, we generate a group of 8 responses using sampling with a temperature of 1.0. We report the average results across three runs with different random seeds for all experiments. All models were trained on a single node with $8 \times$ NVIDIA H100 80GB GPUs. Further details on hyperparameters, such as learning rate and batch size, are provided in Appendix C.

# 7 RESULTS AND ABLATIONS

## 7.1 EXPERIMENTAL RESULT

Table 1 presents the quantitative results on various math benchmarks. For the Qwen2.5-series, we also include the instruct models at the sample scale and R1-Distill models for comparison by following (Liu et al., 2025b). Our proposed reward, PACR, demonstrates significant improvements over the outcome-based reward baseline (+Dr.GRPO) in both its Sparse and Dense setting. This shows that our core method provides a positive inductive bias for improving the reasoning skills of language models.

While the sparse trajectory-level reward, Sparse-PACR, is effective on its own, we observe that Dense-PACR, which provides a more fine-grained reward, consistently achieves better performance. This highlights that enriching the training process with a dense reward signal allows the model to learn from more detailed feedback, leading to further gains in its reasoning capabilities.

---

[1]For the Qwen3-4B model, we set 'enable_thinking=False' to disable its built-in chain-of-thought capabilities, allowing for a direct comparison of how our method versus standard GRPO teaches this capability.

Table 1: **Results on reasoning benchmarks.** We report pass@1 accuracy using temperature $T = 0.0$ across six benchmarks. Both Sparse-PACR and Dense-PACR consistently outperform the Dr.GRPO baseline across all model sizes. † is marked for the score reproduced and other baseline scores are from Liu et al. (2025b). The green colored numbers in the Average column indicate the absolute performance improvement over the Dr.GRPO baseline.

| Base model + Method | AIME25 | AIME24 | AMC | MATH500 | Minerva | OlympiadBench | Average |
|---|---|---|---|---|---|---|---|
| R1-distill-Qwen-1.5B (Gen. length 8k) | 13.3 | 10.0 | 40.9 | 54.6 | 9.2 | 24.1 | 25.4 |
| R1-distill-Qwen-1.5B + Dr.GRPO † | 16.7 | 20.0 | 50.6 | 75.2 | 24.3 | 34.4 | 36.9 |
| R1-distill-Qwen-1.5B + Sparse-PACR | 20.0 | 16.7 | 53.0 | 76.8 | 29.4 | 37.8 | 38.9 +2.0 |
| R1-distill-Qwen-1.5B + Dense-PACR | 20.0 | 20.0 | 56.6 | 78.0 | 26.5 | 38.8 | 40.0 +3.1 |
| Qwen2.5-Math-1.5B | 3.3 | 20.0 | 32.5 | 33.0 | 12.5 | 22.8 | 20.7 |
| R1-Distill-Qwen-1.5B (Gen. length 3k) | 10.0 | 2.5 | 21.7 | 52.2 | 16.3 | 17.3 | 20.0 |
| Qwen2.5-Math-1.5B-Instruct | 10.0 | 10.0 | 48.2 | 74.2 | 26.5 | 40.2 | 34.8 |
| Qwen2.5-Math-1.5B + Dr.GRPO † | 6.7 | 13.3 | 47.0 | 76.8 | 32.3 | 39.0 | 35.8 |
| Qwen2.5-Math-1.5B + Sparse-PACR | 13.3 | 20.0 | 48.4 | 77.4 | 29.4 | 37.8 | 37.7 +1.9 |
| Qwen2.5-Math-1.5B + Dense-PACR | 13.3 | 23.3 | 49.4 | 77.4 | 31.7 | 39.0 | 39.0 +3.2 |
| Qwen2.5-Math-7B | 6.7 | 16.7 | 38.6 | 50.6 | 9.9 | 16.6 | 23.2 |
| SimpleRL-Zero-7B | 6.7 | 26.7 | 60.2 | 78.2 | 27.6 | 40.3 | 39.95 |
| PRIME-Zero-7B | 16.7 | 16.7 | 62.7 | 83.8 | 36.0 | 40.9 | 42.8 |
| OpenReasoner-Zero- 7B @ 3k | 3.3 | 13.3 | 47.0 | 79.2 | 31.6 | 44.0 | 36.4 |
| R1-Distill-Qwen-7B @ 3k | 20.0 | 10.0 | 26.2 | 60.1 | 23.0 | 23.1 | 27.1 |
| Qwen2.5-Math-7B-Instruct | 16.7 | 16.7 | 53.0 | 83.6 | 29.8 | 42.7 | 40.4 |
| Qwen2.5-Math-7B + Dr.GRPO † | 13.3 | 30.0 | 56.6 | 81.8 | 34.6 | 45.2 | 43.6 |
| Qwen2.5-Math-7B + Sparse-PACR | 13.3 | 36.7 | 55.4 | 82.6 | 34.6 | 45.6 | 44.7 +1.1 |
| Qwen2.5-Math-7B + Dense-PACR | 16.7 | 43.3 | 56.1 | 81.9 | 35.6 | 46.1 | 46.6 +3.0 |
| Qwen3-4B | 6.7 | 13.3 | 32.5 | 40.2 | 9.19 | 39.4 | 23.5 |
| Qwen3-4B + Dr.GRPO † | 20.0 | 40.0 | 63.8 | 88.4 | 33.8 | 46.8 | 48.8 |
| Qwen3-4B + Sparse-PACR | 20.0 | 33.3 | 67.5 | 86.2 | 35.3 | 54.4 | 49.4 +0.7 |
| Qwen3-4B + Dense-PACR | 26.7 | 46.7 | 63.4 | 86.8 | 36.0 | 55.0 | 52.4 +3.6 |

Table 2: **Results on Pass@k.** Using temperature $T = 1.0$, Pass@1 (n=16) is calculated as the average accuracy across 16 sampled trajectories, while Pass@16 represents the probability that at least one of the 16 samples is correct. The green colored numbers in the Average column indicate the absolute performance improvement over the Dr.GRPO baseline.

| Base model + Method | Metric | AIME25 | AIME24 | AMC | MATH500 | Minerva | OlympiadBench | Average |
|---|---|---|---|---|---|---|---|---|
| Qwen3-4B | pass@1 (n=16) | 7.9 | 8.9 | 30.7 | 66.5 | 25.2 | 27.0 | 27.7 |
| Qwen3-4B + Dr.GRPO | pass@1 (n=16) | 21.0 | 24.6 | 61.4 | 85.6 | 33.0 | 53.8 | 46.4 |
| Qwen3-4B + Dense-PACR | pass@1 (n=16) | 30.2 | 30.0 | 66.5 | 86.8 | 33.8 | 56.2 | 50.6 +4.2 |
| Qwen3-4B | pass@16 | 16.7 | 33.3 | 48.2 | 82.8 | 37.9 | 41.6 | 43.4 |
| Qwen3-4B + Dr.GRPO | pass@16 | 40.0 | 50.0 | 84.3 | 94.4 | 46.7 | 70.4 | 64.3 |
| Qwen3-4B + Dense-PACR | pass@16 | 46.7 | 56.7 | 84.3 | 94.8 | 48.9 | 71.1 | 67.1 +2.8 |

## 7.2 SAMPLING EFFICIENCY AND INTRINSIC REASONING CAPABILITY

A critical question in RL is whether performance gains stem from genuine reasoning improvement or merely optimized sampling efficiency (i.e., narrowing the distribution around easy solutions at the expense of diversity) (Yue et al., 2025; Kirk et al., 2023; Yu, 2025). To distinguish these effects, we evaluate our models with a positive sampling temperature ($T = 1.0$), reporting two metrics: **Pass@1 (n=16)** as a proxy for *sampling efficiency* (sharpening probability on correct paths), and **Pass@16** as a proxy for *intrinsic capability* (expanding the manifold of solvable problems).

As shown in Table 2, Dense-PACR consistently outperforms the baseline on both fronts. The gain in **Pass@1** confirms improved efficiency, while the concurrent rise in **Pass@16** demonstrates a genuine expansion of reasoning capability.

## 7.3 DISENTANGLING DENSE SUPERVISION FROM TRAINING STABILITY

A potential confounder for PACR's performance is the mitigation of the vanishing advantage problem. In standard GRPO, if all $N$ sampled trajectories share the same outcome (e.g., all incorrect), the group-relative advantage collapses to zero, providing no learning signal. PACR naturally bypasses this issue by assigning continuous, dense rewards ($C_k$) that differentiate trajectories even when final outcomes are identical.

To disentangle the benefits of dense supervision from simple gradient stability, we implemented the **dynamic sampling** strategy from DAPO (Yu et al., 2025) as a baseline. This method resolves the

Table 3: **Comparison with Dynamic Sampling (Stability Baseline).** To isolate the benefit of dense supervision from training stability, we integrated the dynamic sampling strategy from DAPO (Yu et al., 2025). The green colored numbers in the Average column indicate the absolute performance improvement over the Dr.GRPO baseline.

| Base model + Method | AIME25 | AIME24 | AMC | MATH500 | Minerva | OlympiadBench | Average |
|---|---|---|---|---|---|---|---|
| R1-Distill-Qwen-1.5B (Gen. length 8k) | 13.3 | 10.0 | 40.9 | 54.6 | 9.2 | 24.1 | 25.4 |
| + Dr.GRPO | 16.7 | 20.0 | 50.6 | 75.2 | 24.3 | 34.4 | 36.8 |
| + Dr.GRPO + Dynamic Sampling | 10.0 | 20.0 | 60.2 | 79.6 | 28.9 | 40.4 | 39.0 +2.2 |
| + Dense-PACR | 20.0 | 20.0 | 56.6 | 78.0 | 26.5 | 38.8 | 40.0 +3.2 |
| + Dense-PACR + Dynamic Sampling | 16.7 | 26.7 | 56.6 | 80.6 | 25.7 | 37.3 | 40.6 +3.8 |

vanishing advantage by resampling trajectories until the group contains diverse outcomes, such that it maintains the effective batch size across the training.

Table 3 presents the results. While dynamic sampling indeed boosts the Dr.GRPO baseline (raising accuracy from 36.8% to 39.0%), Dense-PACR (40.0%) consistently outperforms this stabilized baseline. Furthermore, combining both methods yields the highest performance (40.6%). This performance gap confirms that stability alone cannot explain the gains; rather, the confidence growth signal ($C_k$) provides necessary directional guidance, steering the model toward better reasoning beyond mere gradient stabilization.

## 7.4 TRAINING CURVE: PACR ACCELERATES EXPLORATION AND IMPROVES CONVERGENCE

Figure 6 illustrates the training dynamics, plotting the average pass@1 accuracy over training steps (left) and the corresponding rate of accuracy improvement (right). The right plot highlights that both PACR variants have a significantly higher rate of improvement compared to the Dr.GRPO baseline, especially during the critical early exploration phase of RL training. As shown on the left, this accelerated learning ultimately allows the PACR methods to converge to a higher final accuracy.

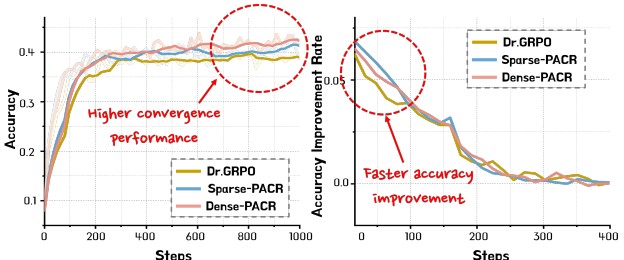

Figure 6: **Training dynamics for Qwen2.5-Math-1.5B.** Average pass@1 accuracy (left) and rate of accuracy improvement (right) during training. PACR methods show a faster initial rate of improvement, accelerating exploration and converging to a higher final performance.

## 7.5 ANALYSIS ON ADVANTAGE FORMULATION: IMPACT OF PENALIZING INTERMEDIATE STEPS

In this section, we analyze the impact of the advantage formulation in our Dense-PACR setting. A crucial design choice is how to normalize the raw discounted returns ($G_{k,\text{dense}}^{(i)}$) into a stable advantage signal. We compare our Min-Max normalization against a widely used Leave-One-Out (LOO) baseline (Ahmadian et al., 2024; Cui et al., 2025).

The key difference is that the LOO baseline centers the returns, which can assign **negative advantages** that penalize steps with below-average confidence gains:

$$A_{k,\text{loo}}^{(i)} = G_{k,\text{dense}}^{(i)} - \text{mean}(\{G_{k,\text{dense}}^{(j)}\}_{j=1,j\neq i}^{N}). \quad (13)$$

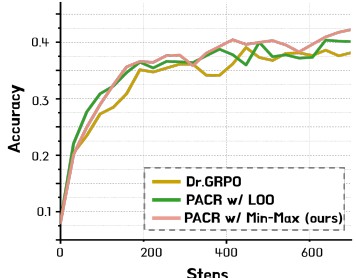

Figure 7: **Advantage Normalization.** Comparison of Min-Max and Leave-One-Out (LOO) for Dense-PACR on Qwen2.5-Math-1.5B.

In contrast, our Min-Max normalization (Eq. 11) scales returns to a $[0, 1]$ range, creating a **purely positive signal** for the reasoning process that only rewards confidence growth.

Figure 7 shows this design choice has a clear impact on the training dynamics. The penalizing nature of the LOO baseline initially accelerates learning by aggressively pruning suboptimal steps, but this

Table 4: **Results on Logical Reasoning Benchmarks.** We report pass@1 accuracy using temperature $T = 0.0$ across two benchmarks.

| Base model + Method | K-K | | | | ZebraLogic | | | | | |
|---|---|---|---|---|---|---|---|---|---|---|
| | ppl6 | ppl7 | ppl8 | Avg. | Small | Medium | Large | X-Large | Grid Acc. | Cell Acc. |
| Qwen-3-4B | 94.0 | 89.0 | 85.0 | 89.3 | 99.4 | 94.3 | 61.0 | 11.5 | 57.8 | 72.7 |
| Qwen-3-4B + Dr.GRPO | 89.0 | 82.0 | 79.0 | 83.8 | 98.4 | 92.9 | 59.5 | 8.5 | 56.11 | 71.1 |
| Qwen-3-4B + PACR | 93.0 | 95.0 | 87.0 | 91.7 | 99.1 | 95.4 | 65.5 | 11.0 | 58.2 | 73.7 |

leads to premature convergence and a performance plateau. Conversely, our non-penalizing Min-Max approach encourages more sustained exploration, ultimately converging to a higher final accuracy. With our method, process-level penalization is avoided; a negative training signal is only applied by the main GRPO reward when the model produces a definitively incorrect final answer.

### 7.6 ANALYSIS ON COMPUTATION OVERHEAD

A natural concern with dense rewards is the computational overhead incurred by calculating $C_k$ at every reasoning step. While these values are computed via batched forward passes, the number of required passes scales linearly with generation length, inevitably increasing the wall-clock time per training iteration compared to the standard sparse reward baseline.

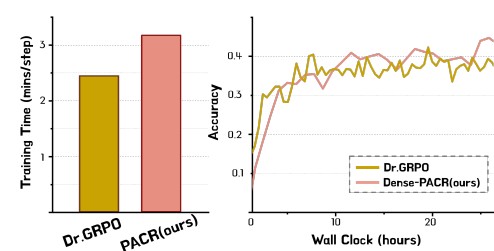

(a) Training Time Comparison per Step  (b) Accuracy vs. Wall Clock

Figure 8: **Computation Overhead on Qwen2.5-Math-1.5B.**

To quantify this trade-off, we compare the training efficiency in Figure 8. As shown in Figure 8-(a), PACR indeed incurs a higher computational cost per step compared to the sparse baseline (Dr.GRPO). However, Figure 8-(b) demonstrates that in terms of **Time-to-Convergence**, PACR is more efficient. When accuracy is analyzed as a function of total wall-clock training time, the PACR curve lies above the baseline. This indicates that the acceleration in learning provided by the dense signal effectively outweighs the overhead of the additional forward passes.

### 7.7 EXPAND TO LOGICAL REASONING

To test the generalization of our method beyond mathematics, we expanded our experimental scope to the domain of logical reasoning. We explicitly trained our models on Knights and Knaves (K-K) train set Xie et al. (2024), utilizing the verifiable version preprocessed by Xie et al. (2025). We evaluate on the test set of K-K and ZebraLogic (Lin et al., 2024) benchmark.

As shown in Table 4, this domain poses a unique challenge for standard RL: the Dr.GRPO baseline exhibits performance degradation compared to the base model (e.g., K-K Average drops from 89.3% to 83.8%), suggesting that sparse rewards are insufficient for credit assignment in brittle logical chains. In contrast, **PACR recovers and exceeds the base model's performance** (e.g., 91.7% on K-K).

## 8 CONCLUSION

In this work, we addressed the limitations of sparse, outcome-based rewards in RLVR by introducing the Progressively Ascending Confidence Reward (PACR), a dense, model-intrinsic signal derived from the model's evolving belief in the ground-truth answer. Through a series of empirical observations and a formal theoretical proof, we validated that confidence growth serves as a powerful inductive bias, effectively constraining the search space to regions richer in logically sound and faithful reasoning paths. Our experiments demonstrated that augmenting GRPO with PACR not only accelerates training but also converges to a higher final performance across multiple reasoning benchmarks, with the fine-grained Dense-PACR variant proving most effective. Ultimately, our work shows that informative, dense rewards for complex reasoning can be effectively extracted from the internal dynamics of the learning policy itself, suggesting a promising direction for creating more effective and reliable methods for fine-tuning the reasoning capabilities of large language models.

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

# A APPENDIX

## A.1 LIMITATIONS AND FUTURE WORK

While our study demonstrates that Progressively Ascending Confidence Reward (PACR) provides a powerful inductive bias for mathematical and logical reasoning, we acknowledge that our current evaluation is primarily confined to natural language Chain-of-Thought (CoT).

**Extension to Multimodal Reasoning.** A direct extension of this work is to investigate the efficacy of the PACR framework in multimodal domains. Visual math problems, for instance, require Vision Language Models (VLMs) to ground reasoning in visual evidence. We hypothesize that the principle of uncertainty reduction applies equally to visual grounding, making this a promising direction for future research.

**Extension to Code-Aided Reasoning.** Furthermore, we note the growing paradigm of **code-aided reasoning**, where models utilize external tools or generate Python code to verify intermediate logic rather than relying solely on natural language (Chen et al., 2022; Gao et al., 2023; Gou et al., 2023; Yu et al., 2024; Yang et al., 2025b). In this domain, applying the standard newline-based segmentation would be suboptimal due to the syntactic density of code. However, the PACR framework is designed to be modular with respect to step granularity. For code-aided tasks, we propose redefining the "reasoning step" as the execution of a functional code block (e.g., the entire content within code delimiters). We posit that the execution of such a block and the retrieval of its output constitutes a single, rigorous event of *uncertainty reduction*. Extending the dense confidence signal to these executable environments represents a high-value direction for future work.

## A.2 BROADER IMPACT

This work introduces a new inductive bias designed to improve the reasoning capabilities of Large Language Models. By leveraging the model's intrinsic confidence dynamics, our method provides fine-grained, step-level supervision without the significant overhead of training separate reward models or requiring manual data annotation. By eliminating the need for external process-reward models or human-annotated datasets, this research significantly lowers the computational and financial barriers to entry for training sophisticated reasoning agents.

## A.3 THE USE OF LLMS

We used LLMs solely for light editing such as correcting grammatical errors and polishing some words. They did not contribute to research ideation, experiments, analysis, or substantive writing. We have reviewed all AI-assisted edits and take full responsibility for the final content of this paper.

## A.4 ETHIC STATEMENT

This research adheres to the highest standards of academic integrity. All existing work is appropriately cited, and this paper does not violate the use of others' work without reference. The experiments conducted do not introduce new datasets or utilize any sensitive data.

# B REASONING SEGMENTATION STRATEGY

A critical prerequisite for any process-based reward framework is the decomposition of the reasoning trajectory $\tau^{(i)}$ into a discrete sequence of steps $\{h_k^{(i)}\}_{k=1}^{K_i}$. The definition of a "step" determines the granularity of credit assignment and directly impacts the stability of the reward signal.

Existing literature in process supervision typically adopts one of three segmentation paradigms:

- **Format-Constrained Segmentation (via SFT):** Some methods rely on Supervised Fine-Tuning (SFT) to enforce rigid output structures, training the model to generate explicit tokens such as "<step>" or "Step $k$:". While this trivializes the parsing process, it introduces a dependency on high-quality, human-annotated process data to bootstrap the format. In this

work, we follow the **DeepSeek-R1-Zero** paradigm (Guo et al., 2025), aiming to incentivize reasoning capabilities directly from the base model via RLVR without relying on extensive supervised cold-start data. Consequently, strategies requiring pre-learned delimiters are incompatible with our training objective.

- **Dynamic Entropy-Based Segmentation:** Recent works have explored leveraging intrinsic uncertainty signals to segment reasoning. For example, Liu et al. (2025a) propose dividing steps at points of high perplexity, hypothesizing that these represent semantic decision boundaries. While theoretically elegant, these methods add computational overhead during training and can be unstable during the early phases of RL when the model's probability distribution is shifting rapidly.

- **Heuristic Delimiter-Based Segmentation:** The most widely adopted approach in the process reward literature (Yang et al., 2025d; Zeng et al., 2025a; Lee et al., 2025) utilizes linguistic heuristics to identify thought boundaries. Common delimiters include newline characters ($\backslash$n) or sentence-terminating punctuation (e.g., ". "). **We adopt this strategy in our work.** Beyond its computational efficiency, this method aligns with the natural syntactic structure of Chain-of-Thought reasoning, where newlines typically signal a transition between logical operations.

## C  TRAINING DETAILS

We present the details of our training configuration as follows. We use a total batch size of 128 and perform one PPO epoch per rollout. The per-device batch size is set to 4 for Qwen2.5-Math-1.5B, and 2 for both Qwen2.5-Math-7B and Qwen3-4B. During rollouts, we use a sampling temperature of 1.0 and generate 8 rollouts per prompt. For optimization, we use the AdamW optimizer (Loshchilov & Hutter, 2017) with a constant learning rate of 1e-6, without warmup or scheduler. The maximum prompt and generation lengths are set to 1024 and 3000 tokens, respectively. For the KL penalty, we set the coefficient $\beta = 0$, effectively deactivating it during training. For the $\lambda_1$, and $\lambda_2$, we search in the range of [1, 0.99, 0.9, 0.8, 0.5] and [0.01, 0.1, 0.2, 0.5], and for the both sparse and dense setting, $\lambda_1$ and $\lambda_2$ are set to 0.9, and 0.1, respectively

## D  PROMPT USED FOR OBSERVATION

To analyze the coherence of the reasoning paths (Observation 2) and the correlation between the large stepwise confidence gain and the pivotal reasoning step (Observation 3) in Section 4.1, we utilize GPT-5 as an evaluator. The prompts used to evaluate the reasoning steps for these respective observations are shown in Figures 9 and 10.

## E  EXAMPLES FOR OBSERVATION 3

This section provides additional qualitative examples that support the central claim of Observation 3. As illustrated by the reasoning trajectories from Qwen2.5-Math-7B (Figure 11) and GPT-OSS-20B (Figure 12, 13 and 14), large positive spikes in the stepwise confidence gain $C_k$ consistently align with pivotal problem-solving steps, such as applying a key formula or executing a critical calculation.

**Discussion: Confidence Saturation in Post-Pivotal Steps.** We also observe a phenomenon we term "confidence saturation." In some trajectories, after a pivotal step yields a massive confidence gain (e.g., $C_k > +2.0$), the immediately subsequent steps often exhibit near-zero gains ($C_{k+1} \approx 0$), even when they represent valid and necessary algebraic manipulations.

While this might initially appear as a failure of the metric (under-rewarding valid steps), we argue that it correctly reflects the information dynamics of reasoning. Once the pivotal insight is established, the remaining uncertainty regarding the final answer drops significantly. Crucially, our advantage formulation utilizes Min-Max normalization rather than Z-score normalization (as discussed in Section 7.5). This design choice ensures that the reward signal remains strictly non-negative ($R \in [0, 1]$). Consequently, these valid post-pivotal steps receive a neutral reward rather than a negative penalty. This prevents the optimization process from actively discouraging necessary execution steps.

**USER**

You are a strict verifier. Given a question, and a proposed thinking process,
assign a LOGIC score from 0-5 for how logically valid the thinking is.

Scoring rubric (integers only):
5 = Fully sound: steps follow logically from the question; no gaps; math/symbol use correct.
4 = Mostly sound: one minor gap/assumption or small imprecision; overall valid.
3 = Mixed: at least one non-trivial gap or unjustified step; partially correct reasoning.
2 = Largely flawed: major gaps, speculative leaps, or misuse of evidence; little support.
1 = Almost entirely illogical: mostly wrong or incoherent reasoning.
0 = Nonsensical/contradictory or unrelated to the question.

Rules:
- Evaluate the thinking itself, not whether the final option is correct.
- Assume minor grammar issues are irrelevant.
- Do not penalize brevity if logically sufficient.
- Refer to visible evidence briefly, but DO NOT reveal your own chain-of-thought.
- Output JSON ONLY with fields: {{"justification": "<<=40 words>", "score": <0-5>}}.

Question: {question}
Proposed thinking:
{rationale}

Output a LOGIC score from 0-5 for how logically valid the thinking is.

Figure 9: **Prompt used to evaluate reasoning quality for Observation 2.**

**USER**

You are a PAIRWISE key-step judge.

Goal: Given a question, the full reasoning trajectory, and its predicted answer, decide which candidate step
(i or j) more causally increases the likelihood of arriving at the given predicted answer.

VALID_IDS = {valid_indices}

Inputs:
- Question: {question}
- Full reasoning trajectory (JSON array of objects {{id, text}}): {steps_json}
- Predicted answer: {predicted_answer}
- Compare: i = {i}, j = {j}

Decision rules:
- Judge only by causal contribution toward the *predicted* answer within the provided steps.
- Prefer the step that adds or applies a decisive constraint, calculation, elimination, or conclusion that moves
the reasoning closer to (or locks in) the predicted answer.
- Do not infer facts or use outside knowledge. Use VALID_IDS only.

Output (STRICT JSON; no extra text):
{{"winner": "i" | "j" | "TIE"}}

Figure 10: **Prompt used to evaulate pairwise comparison for the impactful reasoning steps in Observation 3.**

**Question:** Let $x^8 + 3x^4 - 4 = p_1(x)p_2(x)\dots p_k(x)$, where each non-constant polynomial $p_i(x)$ is monic with integer coefficients, and cannot be factored further over the integers. Compute $p_1(1) + p_2(1) + \dots + p_k(1)$.

**Reasoning:**

$h_1$ | $C_1: +0.15$ | To solver the problem, we need to factor the polynomial $(x^8 + 3x^4 - 4)$ into irreducible factors over the integers and then evaluate the sum of the values of these factors at $(x = 1)$.

$h_2$ | $C_2: -0.10$ | Let's start by factoring the polynomial.

$h_3$ | $C_3: +0.12$ | We can use a substitution to make the polynomial easier to factor.

$h_4$ | $C_4: -0.18$ | Let $(y = x^4)$.

$h_5$ | $C_5: +0.35$ | Then the polynomial becomes $(y^2 + 3y - 4)$.

$h_6$ | $C_6: +1.08$ | We can factor this quadratic polynomial as follows: $y^2 + 3y - 4 = (y - 1)(y + 4)$. Substituting back $(y = x^4)$, we get: $x^8 + 3x^4 - 4 = (x^4 - 1)(x^4 + 4)$. Next, we need to factor $(x^4 - 1)$ and $(x^4 + 4)$ further.

$h_7$ | $C_7: +0.03$ | The polynomial $(x^4 - 1)$ can be factored as: $x^4 - 1 = (x^2 - 1)(x^2 + 1) = (x - 1)(x + 1)(x^2 + 1)$. The polynomial $(x^4 + 4)$ can be factored using the Sophie Germain identity: $x^4 + 4 = (x^2 + 2x + 2)(x^2 - 2x + 2)$. These factors are all irreducible over the integers.

$h_8$ | $C_8: \; 0.00$ | Now, we need to evaluate each factor at $(x = 1)$ and sum the results: $p_1(1) = 1 - 1 = 0, p_2(1) = 1 + 1 = 2$, $p_3(1) = 1^2 + 1 = 2, p_4(1) = 1^2 + 2 \cdot 1 + 2 = 5, p_5(1) = 1^2 - 2 \cdot 1 + 2 = 1$.

$h_9$ | $C_9: -0.01$ | So, the sum is: $0 + 2 + 2 + 5 + 1 = 10$.

Figure 11: **Example trajectory from Qwen2.5-Math-7B for a polynomial factorization problem.** The model tackles the 8th-degree polynomial by reducing it to a quadratic form, then applying systematic factorization to break it down into irreducible integer-coefficient components. The pivotal moment occurs at step $h_6$, where the model executes the core factorization. By successfully factoring the simplified quadratic expression $y^2 + 3y - 4$ and subsequently substituting $x^4$ back in for $y$, the model achieves the central algebraic decomposition of the original polynomial into two more tractable factors ($x^4 - 1$ and $x^4 + 4$). This step represents the main breakthrough required to solve the problem, and the corresponding confidence gain ($C_6 = +1.08$) indicates the model's recognition that the most significant hurdle in the factorization process has been overcome.

**Question:** If $f(x) = ax^4 - bx^2 + x + 5$ and $f(-3) = 2$, then what is the value of $f(3)$?

**Reasoning:**

$h_1$ | $C_1: -2.25$ | We have polynomial $f(x) = ax^4 - bx^2 + x + 5$.

$h_2$ | $C_2: +2.71$ | We know $f(-3) = 2$.

$h_3$ | $C_3: -2.78$ | We need $f(3)$.

$h_4$ | $C_4: +0.12$ | We don't know $a, b$.

$h_5$ | $C_5: +4.40$ | But maybe symmetry? Let's compute $f(-3)$ expression: $a \cdot (-3)^4 - b \cdot (-3)^2 + (-3) + 5$ $= 81a - 9b - 3 + 5 = 81a - 9b + 2$.

$h_6$ | $C_6: +3.96$ | Set equal to 2: $81a - 9b + 2 = 2 \Rightarrow 81a - 9b = 0 \Rightarrow$ divide 9: $9a - b = 0 \Rightarrow b = 9a$.

$h_7$ | $C_7: \; 0.00$ | Now $f(3) = a \cdot 3^4 - b \cdot 3^2 + 3 + 5 = 81a - 9b + 8$.

$h_8$ | $C_8: \; 0.00$ | Substitue $b = 9a$: $81a - 9 \cdot (9a) + 8 = 81a - 81a + 8 = 8$.

$h_9$ | $C_9: \; 0.00$ | So answer 8.

Figure 12: **Example trajectory from GPT-OSS-20B for a polynomial symmetry problem.** The problem appears ostensibly unsolvable due to the unknown coefficients $a$ and $b$. The critical insight emerges across steps $h_5$ and $h_6$, where the model leverages the inherent symmetry of the function's even-powered terms ($ax^4, -bx^2$) and utilizes the given condition $f(-3) = 2$. By evaluating the expression for $f(-3)$ and equating it to 2, the model uncovers the essential relationship between the unknown coefficients ($b = 9a$). This discovery is the pivotal moment that unlocks the entire problem, as it enables the cancellation of the unknown terms in the subsequent calculation of $f(3)$.

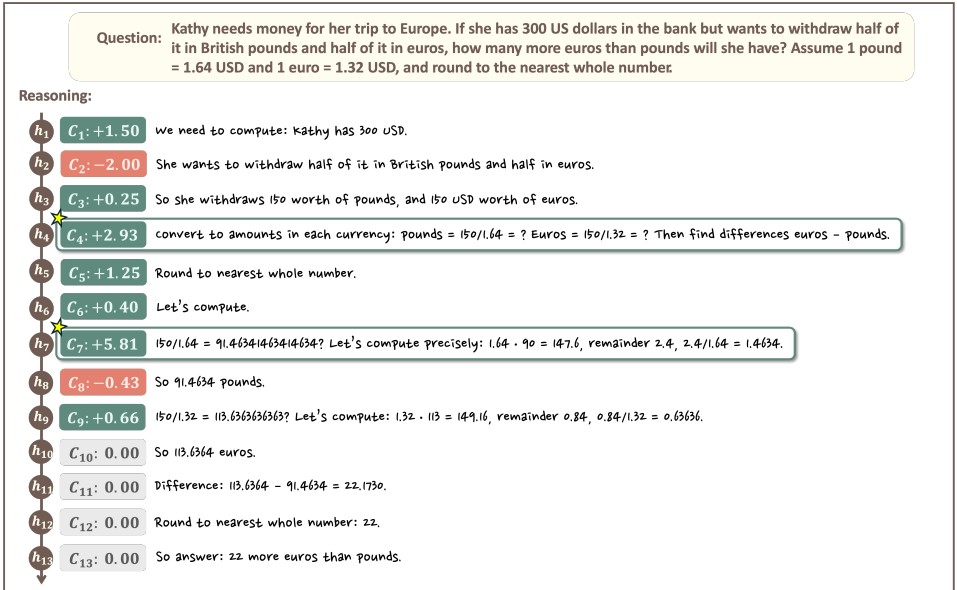

**Question:** The sum of 27 consecutive positive integers is $3^7$. What is their median?

Reasoning:

$h_1$ | $C_1: -3.12$ | We need to find 27 consecutive positive integers whose sum is $3^7$.

$h_2$ | $C_2: +0.12$ | $3^7 = 2187$.

$h_3$ | $C_3: +1.25$ | Sum of 27 consecutive integers: if they are $a, a+1, ..., a+26$.

$h_4$ | $C_4: +1.87$ | Sum $= 27a + (0 + 1 + \cdots + 26) = 27a + \frac{(26 \cdot 27)}{2} = 27a + 351$

$h_5$ | $C_5: +1.75$ | Because sum of 0 to 26 $= \frac{26 \cdot 27}{2} = 351$.

$h_6$ | $C_6: -3.25$ | So $27a + 351 = 2187$.

$h_7$ | $C_7: +6.28$ | Solve: $27a = 2187 - 351 = 1826$.

$h_8$ | $C_8: +5.18$ | $1836/27 =? 27 \cdot 68 = 1836$.

$h_9$ | $C_9: -0.29$ | So $a = 68$.

$h_{10}$ | $C_{10}: +0.33$ | So the numbers are 68 to 94 inclusive.

$h_{11}$ | $C_{11}: 0.00$ | Median of 27 numbers is the 14th number (since 27 odd).

$h_{12}$ | $C_{12}: 0.00$ | Starting at 68, 14th number is $68 + 13 = 81$.

$h_{13}$ | $C_{13}: 0.00$ | So median is 81.

**Figure 13: Example trajectory from GPT-OSS-20B for a consecutive integer median problem.**
The initial phase of this problem involves correctly establishing an algebraic equation for the sum of the integers: $27a + 351 = 2187$. The pivotal moment is a two-step process spanning $h_7$ and $h_8$. In $h_7$, the model transitions from algebraic formulation to the decisive computation to solve for the first integer, $a$, by successfully isolating the term $27a$. This is immediately followed by the critical execution in $h_8$, where the actual division is performed to find the explicit value of $a$. Securing the value of $a$ is the primary breakthrough, as it provides the key to determining all numbers in the sequence and thus the median.

**Question:** Kathy needs money for her trip to Europe. If she has 300 US dollars in the bank but wants to withdraw half of it in British pounds and half of it in euros, how many more euros than pounds will she have? Assume 1 pound = 1.64 USD and 1 euro = 1.32 USD, and round to the nearest whole number.

Reasoning:

$h_1$ | $C_1: +1.50$ | We need to compute: Kathy has 300 USD.

$h_2$ | $C_2: -2.00$ | She wants to withdraw half of it in British pounds and half in euros.

$h_3$ | $C_3: +0.25$ | So she withdraws 150 worth of pounds, and 150 USD worth of euros.

$h_4$ | $C_4: +2.93$ | convert to amounts in each currency: pounds = 150/1.64 = ? Euros = 150/1.32 = ? Then find differences euros - pounds.

$h_5$ | $C_5: +1.25$ | Round to nearest whole number.

$h_6$ | $C_6: +0.40$ | Let's compute.

$h_7$ | $C_7: +5.81$ | 150/1.64 = 91.46341463414634? Let's compute precisely: 1.64 · 90 = 147.6, remainder 2.4, 2.4/1.64 = 1.4634.

$h_8$ | $C_8: -0.43$ | So 91.4634 pounds.

$h_9$ | $C_9: +0.66$ | 150/1.32 = 113.63636363? Let's compute: 1.32 · 113 = 149.16, remainder 0.84, 0.84/1.32 = 0.63636.

$h_{10}$ | $C_{10}: 0.00$ | So 113.6364 euros.

$h_{11}$ | $C_{11}: 0.00$ | Difference: 113.6364 − 91.4634 = 22.1730.

$h_{12}$ | $C_{12}: 0.00$ | Round to nearest whole number: 22.

$h_{13}$ | $C_{13}: 0.00$ | So answer: 22 more euros than pounds.

**Figure 14: Example trajectory from GPT-OSS-20B for a currency exchange problem.**
This reasoning trajectory features two pivotal moments. First, step $h_4$ serves as a critical planning phase, where the model correctly formulates the computational roadmap required for the solution: two currency conversions via division, followed by a subtraction. This demonstrates a comprehensive understanding of the problem's logic. The second, more significant pivotal moment occurs at the execution phase in step $h_7$, where the model accurately performs the first of the two required divisions. Successfully clearing this key computational hurdle provides the model with high confidence ($C_7 = +5.81$) that its strategy is effective and the path to the final answer is now clear.

## F  LABEL ACQUISITION FOR STEP QUALITY ANALYSIS

To empirically validate that confidence growth accurately tracks fine-grained reasoning quality (Section 4.1), we required a set of ground truth quality labels for individual reasoning steps. We constructed a high-quality annotated set using a state-of-the-art Large Language Model as a programmatic annotator.

### F.1  ANNOTATION SETUP

We sampled a diverse set of reasoning trajectories generated by the `Qwen2.5-Math-7B`, `GPT-oss-20B`, and `Qwen3-4B` models from the MATH benchmark (Hendrycks et al.) test set. Each trajectory was first segmented into discrete steps following the delimiter-based rules described in Section 3. Subsequently, we utilized `gpt-5.1` (snapshot `2025-11-13`) to classify every individual step's quality within these trajectories.

To define the quality criteria, we adopted the taxonomy established in previous process supervision literature Lightman et al. (2023), adapting it to capture the granularity of information gain. The model was provided with the relevant context (Question, Final Ground Truth, Reasoning History up to the current step) and the specific Candidate Step. The exact system instruction provided to the annotator is as follows:

---

**Prompt: System Instruction for Step Quality Annotation**

```
Role:  You are grading ONE intermediate step in a student's solution
to a math problem.
Rate the quality of the CURRENT step using exactly these labels:
```

- GREAT:  A strong step that a good math student might take.
  It clearly moves the solution forward or is a reasonable
  attempt to make mathematical progress, even if it's not
  perfectly optimal.
- OKAY:  Plausible but low-value.  It may restate or lightly
  rephrase things, check an obvious detail, or otherwise fail
  to add real insight or progress, but it is not clearly wrong
  or misleading.
- BAD:  Confidently wrong, off-topic, incoherent, or clearly
  leading the solution toward a dead end; OR technically
  correct but explained so poorly that a typical student could
  not follow it.

```
Context Rule:  Always judge the current step in the context of the
problem and the previous steps.
Output Format:  Respond with STRICT JSON only, of the form:
```

- rating ("Great" | "Okay" | "Bad")
- reason ("short explanation")

```
Do not include any extra keys or any text outside the JSON.
```

---

```
Input Template:
Problem: {question}
Ground-truth final answer (if available): {gt_answer}
Model's final answer (if available): {model_final_answer}
Reasoning so far (steps 1..k, including the current step):
{steps_upto_str}
Current step to rate (this is the LAST step above):
{current_step_text}
Now output JSON only.
```

---

**Handling Error Propagation (First-Error Truncation).**  Consistent with the labeling method used in (Lightman et al., 2023), we adopt a "first-error" truncation strategy. Since language models are autoregressive, every reasoning step is conditioned on the entire preceding history. Consequently, once a step is labeled BAD (indicating a logical error or hallucination), the validity of all subsequent

steps is compromised by the flawed context. To avoid the ambiguity of grading reasoning based on false premises, we terminate annotation immediately upon encountering the first `BAD` step; all subsequent steps in that trajectory are excluded from our analysis.

## F.2 VALIDATION OF LABEL QUALITY

To verify the reliability of this automated annotation, we performed a rigorous inter-annotator agreement study:

1. **Human Inter-Annotator Agreement:** Two human experts (graduate students in mathematics/computer science) independently annotated a random subset of 100 steps. They achieved a Cohen's Kappa of $\kappa = 0.76$, indicating that the distinction between Great, Okay, and Bad steps is well-defined and unambiguous to humans.

2. **Model-Human Alignment:** We compared the primary `gpt-5.1` annotations against the human consensus on the same subset. The model achieved a Kappa score of $\kappa = 0.72$ (Table 5). This substantial alignment confirms that the model effectively acts as a reliable proxy for human judgment, correctly adhering to the strict definitions provided in the prompt.

Table 5: **Inter-Annotator Agreement Scores.** The substantial agreement ($\kappa > 0.7$) validates that the labels are reliable proxies for reasoning quality.

| Comparison Pair | Metric Interpretation | Cohen's $\kappa$ |
|---|---|---|
| Human Expert 1 vs. Expert 2 | Task Definition Quality | 0.76 |
| GPT-5.1 vs. Human Consensus | Proxy Reliability | 0.72 |

## G THEORETICAL MOTIVATION FOR GROUND-TRUTH CONFIDENCE GROWTH AS A PROCESS REWARD

Building on our empirical findings, we provide the theoretical motivation for using confidence growth as a process reward. We demonstrate that maximizing this reward mathematically aligns the model's reasoning process with a superior "oracle" distribution conditioned on the correct answer.

**The Oracle Policy: Conditioning on the correct answer as a superior policy** We define the *oracle policy*, $\pi_{\text{oracle}}$, as the model's generative process when conditioned on the ground-truth answer $Y_{\text{gt}}$:

$$\pi_{\text{oracle}}(h_k) \triangleq \pi_\theta(h_k \mid q, Y_{\text{gt}}, H_{<k})$$

A critical premise is that $\pi_{\text{oracle}}$ represents a "better" policy than the training policy $\pi_\theta$ (which generates steps without access to the answer). Framing the policy conditioned on the correct answer as a superior objective aligns with recent works (Zelikman et al., 2022; Wang et al., 2025), which uses a ground-truth conditioned policy to sample good reasoning steps.

**Confidence Gain as Implicit Imitation** We now show how our proposed reward, confidence gain ($C_k$), leverages this oracle distribution. Recall the definition of $C_k$ from Eq. 6:

$$C_k = \log \pi_\theta(Y_{\text{gt}} \mid q, H_{\leq k}) - \log \pi_\theta(Y_{\text{gt}} \mid q, H_{<k})$$

Applying Bayes' theorem to the first term, $\pi_\theta(Y_{\text{gt}} \mid q, h_k, H_{<k})$, allows us to express $C_k$ as the log-likelihood ratio between the oracle and the standard policy:

$$C_k = \log \frac{\pi_\theta(h_k \mid q, Y_{\text{gt}}, H_{<k}) \cdot \pi_\theta(Y_{\text{gt}} \mid q, H_{<k})}{\pi_\theta(h_k \mid q, H_{<k}) \cdot \pi_\theta(Y_{\text{gt}} \mid q, H_{<k})} = \log \frac{\pi_{\text{oracle}}(h_k)}{\pi_\theta(h_k)} \tag{14}$$

During training, the model generates steps $h_k$ according to its current policy $\pi_\theta$ and aims to maximize the expected reward $\mathbb{E}[C_k]$. By substituting Eq. 14 into the objective function $J(\theta)$, we obtain:

$$J(\theta) = \mathbb{E}_{h_k \sim \pi_\theta}[C_k]$$

$$= \mathbb{E}_{h_k \sim \pi_\theta}\left[\log \frac{\pi_{\text{oracle}}(h_k)}{\pi_\theta(h_k)}\right]$$

$$= -D_{KL}(\pi_\theta \parallel \pi_{\text{oracle}}).$$

This derivation demonstrates that maximizing the confidence gain is equivalent to minimizing the KL divergence between the current policy $\pi_\theta$ and the oracle policy.

## H  STANDARD DEVIATION FOR EXPERIMENTS

Table 6: **Standard Deviation across 3 random seeds.** We report the standard deviation of pass@1 accuracy using temperature $T = 0.0$ across six benchmarks. Lower values indicate more stable performance. Baseline Dr.GRPO shows higher variance due to training instability, whereas PACR methods demonstrate consistently lower variance.

| Base model + Method | AIME25 | AIME24 | AMC | MATH500 | Minerva | OlympiadBench | Average |
|---|---|---|---|---|---|---|---|
| R1-distill-Qwen-1.5B (Gen. length 8k) | - | - | - | - | - | - | - |
| R1-distill-Qwen-1.5B + Dr.GRPO † | 2.1 | 1.8 | 2.5 | 1.2 | 1.5 | 1.9 | 1.8 |
| R1-distill-Qwen-1.5B + Sparse-PACR | 0.8 | 0.6 | 1.1 | 0.5 | 0.7 | 0.9 | 0.8 |
| R1-distill-Qwen-1.5B + Dense-PACR | 0.5 | 0.8 | 0.9 | 0.6 | 0.8 | 0.7 | 0.7 |
| Qwen2.5-Math-1.5B | - | - | - | - | - | - | - |
| R1-Distill-Qwen-1.5B (Gen. length 3k) | - | - | - | - | - | - | - |
| Qwen2.5-Math-1.5B-Instruct | - | - | - | - | - | - | - |
| Qwen2.5-Math-1.5B + Dr.GRPO † | 1.8 | 2.1 | 2.4 | 1.1 | 1.6 | 1.7 | 1.8 |
| Qwen2.5-Math-1.5B + Sparse-PACR | 0.8 | 0.9 | 1.0 | 0.6 | 0.8 | 0.7 | 0.8 |
| Qwen2.5-Math-1.5B + Dense-PACR | 0.6 | 0.7 | 0.8 | 0.5 | 0.6 | 0.8 | 0.7 |
| Qwen2.5-Math-7B | - | - | - | - | - | - | - |
| SimpleRL-Zero-7B | - | - | - | - | - | - | - |
| PRIME-Zero-7B | - | - | - | - | - | - | - |
| OpenReasoner-Zero- 7B @ 3k | - | - | - | - | - | - | - |
| R1-Distill-Qwen-7B @ 3k | - | - | - | - | - | - | - |
| Qwen2.5-Math-7B-Instruct | - | - | - | - | - | - | - |
| Qwen2.5-Math-7B + Dr.GRPO † | 2.0 | 2.5 | 2.2 | 1.4 | 1.8 | 2.1 | 2.0 |
| Qwen2.5-Math-7B + Sparse-PACR | 0.9 | 0.8 | 1.2 | 0.6 | 0.9 | 0.8 | 0.9 |
| Qwen2.5-Math-7B + Dense-PACR | 0.7 | 0.8 | 1.0 | 0.5 | 0.8 | 0.9 | 0.8 |
| Qwen3-4B | - | - | - | - | - | - | - |
| Qwen3-4B + Dr.GRPO † | 2.5 | 2.8 | 2.4 | 1.2 | 1.9 | 2.3 | 2.2 |
| Qwen3-4B + Sparse-PACR | 1.1 | 1.2 | 1.4 | 0.7 | 1.0 | 1.3 | 1.1 |
| Qwen3-4B + Dense-PACR | 0.9 | 0.8 | 1.1 | 0.6 | 0.9 | 1.0 | 0.9 |

Table 7: **Comparison with Chunk-level PACR Experiment.** The green colored numbers in the Average column indicate the absolute performance improvement over the Dr.GRPO baseline.

| Base model + Method | AIME25 | AIME24 | AMC | MATH500 | Minerva | OlympiadBench | Average |
|---|---|---|---|---|---|---|---|
| R1-Distill-Qwen-1.5B (Gen. length 8k) | 13.3 | 10.0 | 40.9 | 54.6 | 9.2 | 24.1 | 25.4 |
| + Dr.GRPO | 16.7 | 20.0 | 50.6 | 75.2 | 24.3 | 34.4 | 36.8 |
| + Dense-PACR | 20.0 | 20.0 | 56.6 | 78.0 | 26.5 | 38.8 | 40.0 +3.2 |
| + Dense-PACR with chunk 2 | 13.3 | 20.0 | 56.6 | 80.8 | 26.1 | 36.4 | 39.7 +2.9 |
| + Dense-PACR with chunk 4 | 20.0 | 16.7 | 52.8 | 78.6 | 26.8 | 37.4 | 38.7 +1.9 |

# I   CHUNK-LEVEL PACR EXPERIMENT

As discussed in Section 7.5, a natural concern with PACR is the computational overhead incurred by calculating $C_k$ at every reasoning step, which requires additional forward passes during the rollout phase. To mitigate this, we investigated a Chunk-Level PACR strategy, where adjacent reasoning steps are aggregated into larger chunks, and the reward is computed only at the end of each chunk (i.e., every $k$ steps). This linearly reduces the number of required forward passes by a factor of $k$.

Table 7 presents the results of this ablation on the R1-Distill-Qwen-1.5B model. We observe a clear trade-off between signal density and computational efficiency. Specifically, aggregating every two steps ($k = 2$) results in an average accuracy of **39.7%**. This performance is **effectively on-par** with the fully dense baseline (Dense-PACR, 40.0%), showing only a marginal decline while halving the reward computation cost. Increasing the chunk size further to $k = 4$ leads to a slightly larger drop to 38.7%, likely due to the dilution of the training signal over longer intervals. However, critically, this performance remains significantly higher than the standard Dr.GRPO baseline (36.8%). This indicates that PACR **retains its efficacy** even with coarser step granularity, offering a practical trade-off between computational cost and signal density.

