# OpenReview forum: "PACR: Progressively Ascending Confidence Reward for LLM Reasoning"
_ICLR.cc/2026/Conference — Submitted to ICLR 2026_

### Official Review · Reviewer_1VLi · 2025-10-15

**Soundness:** 1
**Presentation:** 3
**Contribution:** 2
**Rating:** 2
**Confidence:** 4

**Summary:**

This paper proposes PACR (Progressively Ascending Confidence Reward), a dense reward signal for training LLM reasoning via reinforcement learning. The core idea is to reward intermediate reasoning steps based on whether they increase the model's probability of the ground-truth answer. The authors provide empirical observations and theoretical justification for this approach, implementing both sparse (trajectory-level) and dense (step-level) variants.

**Strengths:**

Conceptual clarity: The idea of using internal confidence dynamics as a reward shaping signal is intuitive and well-motivated.

Implementation simplicity: PACR does not require an external reward model, making it lightweight and practical.

Empirical validation: The paper includes quantitative results on multiple math reasoning benchmarks (MATH500, AIME, AMC, etc.) showing some performance gains.

**Weaknesses:**

## Flawed Theoretical FoundationProposition 1 is trivial and doesn't validate the method's utility:

The proposition proves that E[C_k] ≥ 0 when h_k is sampled from π_θ(·|q, Y_gt, H_<k) (the "oracle policy").
However, during actual RL training, steps are sampled from π_θ(·|q, H_<k) without access to Y_gt.
The proof merely shows that conditioning on the answer increases confidence in that answer—this is tautological.
The critical gap: the paper never establishes that actually sampled steps (without Y_gt conditioning) will exhibit this property.
This undermines the entire theoretical motivation for the method.

## Weak and Inconsistent Empirical Results

Table 1 shows marginal and unstable improvements:

- Many improvements are within noise margins (e.g., +0.6, +0.8 on MATH500)
- I admire the authors' frankness in posting those results. Several datasets show degradations (e.g., Minerva: -2.9 for Sparse-PACR on 1.5B; AMC: -1.2 for Sparse-PACR on 7B)
- The average improvements (≤3.0 points) are modest given the added computational complexity
- No statistical significance testing provided despite claiming results over 3 seeds

## Circular Reasoning in Observational Studies

Observation 1 (Section 4.1) lacks proper controls:

- Finding that correct trajectories have more positive C_k steps could simply mean: correct reasoning leads to correct answers (obvious)
- No analysis controlling for confounding factors (e.g., trajectory length, model confidence calibration)
- Causality is unclear: does confidence growth cause correctness, or does correctness cause confidence growth?

Observation 2's methodology is questionable:

- Using GPT-5 to judge "logical coherence" introduces significant subjectivity and potential bias
- The distinction between "coherent" and "spurious" reasoning that reaches correct answers is ill-defined
- High confidence in a spurious path might actually indicate a model failure rather than validate the reward signal

**Questions:**

Why not compare against actual trained process reward models?

Can you show results with statistical significance tests?

What happens when the model is highly confident in the wrong answer—does PACR still provide useful signal?

Compare against recent dense-reward or implicit-reward baselines (e.g., ConfPO, TLCR, DAPO, DeepSearch).

---

> ### Author Response · Authors · 2025-11-28
> **Response to Reviewer 1VLi (1/6)**
>
> We sincerely appreciate your time and constructive comments, which improve our paper. We address your concerns as follows.
>
> ---
>
> > **[Weakness 1]**  Proposition 1 validates the method only when sampling from the oracle policy, whereas training samples from the current policy $\pi_\theta$. The reviewer suggests this undermines the theoretical motivation.
>
> - We thank the reviewer for the feedback regarding our theoretical motivation. We agree that our original formulation, which analyzed properties under the oracle policy, did not fully address the dynamics of the actual training process where steps are sampled from the student policy $\pi_\theta$.
>
> - Accordingly, we have reformulated our theoretical motivation to derive the objective function directly under the training policy $\pi_\theta$.
>
>     - Similar to our original theoretical motivation, we define the *oracle policy*, $\pi_{\text{oracle}}$, as the model's generative process when conditioned on the ground-truth answer $Y_{\text{gt}}$:
>     $$\pi_{\text{oracle}}(h_k) \triangleq \pi_\theta(h_k \mid q, Y_{\text{gt}}, H_{<k})$$
>
>     - Here, instead of relying on the oracle policy assumption, we frame $\pi_{\text{oracle}}$ as representing a "better" policy than the training policy $\pi_{\theta}$ (which generates steps without access to the answer). This perspective aligns with recent works [1,2], which uses ground-truth conditioned policy to generate reasoning steps for training.
>
>     - We demonstrate that maximizing the expected confidence gain $J(\theta)$ is mathematically equivalent to minimizing the KL divergence between the current policy and the superior oracle policy (due to Bayes Rule. Details are in `Appendix G Eq.14`):
>
>         $$J(\theta) = E_{h_k \sim \pi_{\theta}} [C_k] = E_{h_k \sim \pi_{\theta}} \left[ \log \frac{\pi_{\text{oracle}}(h_k)}{\pi_{\theta}(h_k)} \right] = - D_{KL}(\pi_{\theta} \ \| \ \pi_{\text{oracle}}) $$
>
>
> - This derivation provides a concrete theoretical motivation for our proposed method: it proves that by maximizing this reward, the model implicitly imitates the superior reasoning process of the ground-truth conditioned model.
>
> - Additionally, we have moved this theoretical motivation (`Section 4.2`) to `Appendix G`. This allows the main text to focus on the empirical validation and causal analysis of the reward signal, which we agree are the core contributions of the paper.
>
> ---
>
> ### Reference
>
> [1] Zelikman  et al., "Star: Self-taught reasoner bootstrapping reasoning with reasoning." 2024.
>
> [2] Wang et al., "Think or Not? Selective Reasoning via Reinforcement Learning for Vision-Language Models." 2025.

---

> ### Author Response · Authors · 2025-11-28
> **Response to Reviewer 1VLi (2/6)**
>
> > **[Weakness 2]** Concerns regarding the magnitude and stability of the empirical results.
>
> - We thank the reviewer for highlighting the need for robust empirical validation. We agree that verifying the stability of performance gains is critical for RL methods. In response, we have significantly expanded our evaluation scope to demonstrate the consistency of our method.
>
> - **Consistency of Improvements:** Regarding the magnitude of gains, we emphasize that **Dense-PACR** is our primary proposed method. If we look at `Table 1`, while the Sparse-PACR variant exhibits slightly higher variance, Dense-PACR consistently achieves meaningful performance gains across all experimental settings on the Average score. This confirms the robustness of the dense supervision signal compared to sparse alternatives.
>
> - **Expanded Model and Benchmark Scope:** To further prove robustness across different settings, we added a new base model (**R1-Distill-Qwen-1.5B**) and an additional reasoning benchmark (**AIME 2025**) to our main results (`Table 1`).
>     - Even on the highly optimized R1-Distill model, Dense-PACR achieves a **+3.1%** absolute average gain over the Dr.GRPO baseline.
>
>
> - **Evaluation of Distributional Robustness (Pass@k):** To assess stability under stochastic generation, we expanded our analysis in `Section 7.2` (`Table 2`) to include **Pass@1 (n=16)** and **Pass@16** metrics sampled at a higher temperature ($T=1.0$).
>     - Dense-PACR improves both metrics significantly on the Qwen3-4B model: achieving a **+4.2%** gain in sampling efficiency (Pass@1, n=16) and a **+2.8%** gain in intrinsic capability (Pass@16).
>     - These results demonstrate that PACR provides concrete benefits in generating diverse correct reasoning paths.
>
> - **Statistical Significance:** We include the standard deviation for the `Table1` in `Appendix H` for statistical significnace.
>
> - **Generalization to Logical Reasoning:** Finally, to demonstrate that our method's effectiveness extends beyond mathematics, we conducted additional experiments on **Logical Reasoning** tasks. As detailed in `Section 7.7 (Table 4)`, we observe a critical failure mode for the standard outcome-based baseline (Dr.GRPO), which **actually degrades performance** compared to the untrained base model on logical benchmarks. In contrast, Dense-PACR **recovers and exceeds** the base model's performance.

---

> ### Author Response · Authors · 2025-11-28
> **Response to Reviewer 1VLi (3/6)**
>
> > **[Weakness 3]** Concerns about observational studies: observation 1 shows circular reasoning/is obvious, and causality between confidence growth and correctness is unclear. For observation 2, it is uncertain whether the introduction of GPT-5 to label the reasoning ("coherent" vs. "spurious") is reliable/well-defined.
>
> - We thank the reviewer for their critical assessment of our observational methodology. We address the concern regarding circularity and labeling reliability separately below.
>
>      - Clarification on "Circular Reasoning" (Human vs. Model Cognition): We respectfully posit that the relationship between reasoning progress and confidence growth is **not tautological** in the context of Large Language Models (LLMs).
>          - **Human Intuition:** While it is intuitive for *humans* to experience reduced uncertainty as they near a solution, LLMs are probabilistic systems without inherent cognitive states. It is not obvious *a priori* whether their internal probability mass shifts monotonically during correct reasoning steps.
>          - **Scientific Value:** Our observation confirms that LLMs *do* exhibit this human-like "uncertainty reduction" process. This is a non-trivial empirical finding regarding model alignment, rather than a circular definition.
>      - **Causality:** Moreover, to investigated the causality, we have replaced the original observation2 with a new beam search experiment into the revised `Section 4.1` (see Paragraph b: *`Causal Utility: Does guiding generation with this bias improve accuracy during inference?`*).
>         - Specifically, we implemented a **step-wise Best-of-N search strategy** (equivalent to beam search with width 1). At each reasoning step, we sampled $N=8$ candidates and selected the path maximizing the raw confidence gain ($C_k$).
>         - Crucially, this setup uses the **pure confidence gain as a process reward signal** without the outcome mixing or min-max normalization used during RL training, directly testing the reviewer's hypothesis.
>
>         - As shown in the added `Figure 2-(b)` of `Section 4.1`, the PACR-guided search during inference significantly outperforms standard sampling baselines across multiple temperatures ($T=0.8, 0.9, 1.0$).
>
>
>     - Validation of GPT-5 Label Quality (`Appendix F`): To address the reviewer's concern regarding the subjectivity of GPT-5 annotations. we have conducted a rigorous **Inter-Annotator Agreement Study**. We have added these results to `Appendix F.2`
>
> 	    - **Human-Human Agreement ($\kappa = 0.76$):** Two domain experts (graduate students in Math/CS) independently annotated 100 random steps. The substantial Cohen’s Kappa score demonstrates that our definitions of *Great*, *Okay*, and *Bad* are well-defined and unambiguous to humans.
> 	    - **Model-Human Alignment ($\kappa = 0.72$):** We compared the GPT-5.1 annotations against the human consensus. The high alignment score confirms that the model acts as a reliable proxy for expert human judgment, strictly adhering to the established definitions rather than hallucinating label criteria.

---

> ### Author Response · Authors · 2025-11-28
> **Response to Reviewer 1VLi (4/6)**
>
> > **[Q1]** Why not compare against actual trained process reward models?
>
> - We respectfully clarify that our exclusion of trained PRMs is based on a difference in **data regimes**. Standard PRMs rely on **process supervision**, requiring massive datasets of human or GPT-annotated reasoning steps. In contrast, PACR operates in the **outcome supervision** regime (RLVR), utilizing *only* the final ground truth signal. Comparing PACR directly to a PRM would largely measure the value of the additional annotated data rather than the efficacy of the method itself. Therefore, we benchmark against baselines that also operate without process labels to ensure a fair comparison within the same data constraints.
>
> > **[Q2]** Can you show results with statistical significance tests?
>
> - We have added the standard deviation for `Table1` in `Appendix H` in the revised manuscript.

---

> ### Author Response · Authors · 2025-11-28
> **Response to Reviewer 1VLi (5/6)**
>
> > **[Q3]** What happens when the model is highly confident in the wrong answer—does PACR still provide useful signal?
>
> - We think PACR can provide an useful signal even in this scenario.
> - It is important to distinguish between the model's confidence in its *generated prediction* versus the metric used by PACR, which is the probability assigned to the **ground-truth answer ($Y_{gt}$)**.
> - Because $C_k$ is calculated based on $Y_{gt}$, the mechanism works as follows:
>     - Probability Shift: If the model becomes highly confident in a *wrong* answer, the probability mass shifts towards that incorrect token.
>     - Ground-Truth Decay: Consequently, the probability assigned to the ground-truth answer, $\pi(Y_{gt})$, decreases (or becomes negligible).
>     - Negative Reward: Since $C_k$ measures the *gain* in ground-truth probability, a decrease in $\pi(Y_{gt})$ results in a **negative value** for $C_k$.
> - Therefore, when the model is confident in a wrong answer, PACR correctly identifies this as a reasoning failure.

---

> ### Author Response · Authors · 2025-11-28
> **Response to Reviewer 1VLi (6/6)**
>
> > **[Q4]** Compare against recent dense-reward or implicit-reward baselines
>
> - **Comparison with Dynamic Sampling (DAPO):** To address the concern regarding baseline diversity, we expanded our comparison to include a stability-oriented baseline. Specifically, we integrated the **dynamic sampling strategy** from DAPO, which is explicitly designed to solve the vanishing advantage problem by maintaining a consistent effective batch size. As detailed in the added `Section 7.3 (Table 3)`, this comparison allows us to disentangle the source of our gains.
> 	- We observe that while dynamic sampling improves the Dr.GRPO baseline (raising average accuracy from **36.8%** to **39.0%**), **Dense-PACR (40.0%) consistently outperforms it**. Furthermore, combining both methods yields the highest performance (**40.6%**). This confirms that the performance boost stems from the **informative quality of the dense signal** ($C_k$) itself, providing directional guidance, rather than solely from improved training stability.

---

### Official Review · Reviewer_8r9w · 2025-11-01

**Soundness:** 3
**Presentation:** 4
**Contribution:** 2
**Rating:** 4
**Confidence:** 4

**Summary:**

This paper proposes Progressively Ascending Confidence Reward, called PACR, to improve LLM reasoning inside the RL with verifiable reward setting. The key idea is simple. While the model writes a chain of thought, at each step the authors compute how much the log probability of the true answer increases, and they use this confidence gain as a dense, model intrinsic reward. They give two variants, Sparse PACR at trajectory level and Dense PACR at step level, and combine them with GRPO under RLVR. On three Qwen math models, the paper reports faster exploration and small but consistent gains on AIME 2024, AMC 2023, MATH500, Minerva Math, and OlympiadBench.

**Strengths:**

1. The paper gives a very clear inductive bias: along a good reasoning path, confidence in the ground truth should tend to go up. Turning this into a dense reward that needs no extra reward model is practical and clean.
2. The paper checks three open models and five math benchmarks. The main table shows Dense PACR improves average pass@1 over a strong Dr GRPO baseline, for example +2.5 on the 1.5B model and +3.0 on the 7B model.

**Weaknesses:**

1. Experiments only cover math datasets. Many recent results also evaluate general reasoning and code. It is not clear if PACR transfers beyond numeric answers or beyond tasks where the final answer is exactly verifiable, for example, long form QA or proofs with multiple valid forms. Comparison to broader settings in R1-style training or DAPO-like systems would strengthen the claim.
2. The proof shows that the expected confidence gain is non-negative when steps are drawn from the ground truth conditioned policy. In practice, training never samples from that oracle. This gives a nice intuition but a weak guarantee for the real policy, and it leaves open how often confidence will increase under noisy steps. A discussion that connects the oracle gap to measured gains would help.
3. Because the reward comes from the model’s own probability of the final answer, the model may learn to raise that probability early by adding hints or formatting patterns that correlate with the answer tokens, without improving reasoning quality. PRM works try to avoid such reward hacking with different credit assignment, for example, min form credit or verifier designs that measure true progress. A safety check or a negative control would be useful.

**Questions:**

See weaknesses.

---

> ### Author Response · Authors · 2025-11-28
> **Response to Reviewer 8r9w (1/3)**
>
> We sincerely appreciate your time and constructive comments, which improve our paper. We address your concerns as follows.
>
> ---
>
> > **[Q1]** Experiments only cover math datasets. Many recent results also evaluate general reasoning and code. It is not clear if PACR transfers beyond numeric answers or beyond tasks where the final answer is exactly verifiable, for example, long form QA or proofs with multiple valid forms. Comparison to broader settings in R1-style training or DAPO-like systems would strengthen the claim.
>
> - We thank the reviewer for highlighting the importance of evaluating generalization beyond mathematical domains. We agree that verifying the method's effectiveness on broader reasoning tasks and against more robust training setups is essential.
>
> - We have expanded our experiments in the revised manuscript as follows.
>
> - **Generalization to Logical Reasoning:** To demonstrate that our method's effectiveness extends beyond mathematics, we conducted additional experiments on **Logical Reasoning** tasks. As detailed in `Section 7.7 (Table 4)`, we observe a critical failure mode for the standard outcome-based baseline (Dr.GRPO), which **actually degrades performance** compared to the untrained base model on logical benchmarks. In contrast, PACR **recovers and exceeds** the base model's performance.
>
> - **Comparison with Dynamic Sampling (DAPO):** To address the concern regarding baseline diversity, we expanded our comparison to include a stability-oriented baseline. Specifically, we integrated the **dynamic sampling strategy** from DAPO, which is explicitly designed to solve the vanishing advantage problem by maintaining a consistent effective batch size. As detailed in the added `Section 7.3 (Table 3)`, this comparison allows us to disentangle the source of our gains.
> 	- We observe that while dynamic sampling improves the Dr.GRPO baseline (raising average accuracy from **36.8%** to **39.0%**), **Dense-PACR (40.0%) consistently outperforms it**. Furthermore, combining both methods yields the highest performance (**40.6%**). This confirms that the performance boost stems from the **informative quality of the dense signal** ($C_k$) itself, providing directional guidance, rather than solely from improved training stability.
>
> - **Deepened Evaluation Metrics (Pass@k Analysis):** To ensure our results reflect genuine reasoning improvements rather than just mode collapse, we expanded our evaluation metrics. As detailed in the new `Section 7.2` (`Table 2`), we analyzed **Pass@1 (n=16)** and **Pass@16** with temperature set to $T=1.0$. We found that PACR improves both, confirming that PACR demonstrates a genuine expansion of reasoning capability.
>
> - **Scope regarding Verifiability:** We acknowledge the reviewer's point regarding "tasks where the final answer is exactly verifiable." Our current formulation of PACR relies on a known ground truth ($Y_{gt}$) to compute the confidence gain. Therefore, extending this to open-ended Long-Form QA (where no single $Y_{gt}$ exists) is indeed a limitation of the current verifiable-reward framework.

---

> ### Author Response · Authors · 2025-11-28
> **Response to Reviewer 8r9w (2/3)**
>
> > **[Q2]** The proof shows that the expected confidence gain is non-negative when steps are drawn from the ground truth conditioned policy. In practice, training never samples from that oracle. This gives a nice intuition but a weak guarantee for the real policy, and it leaves open how often confidence will increase under noisy steps. A discussion that connects the oracle gap to measured gains would help.
>
>
> - We thank the reviewer for the feedback regarding our theoretical motivation. We agree that our original formulation, which analyzed properties under the oracle policy, did not fully address the dynamics of the actual training process where steps are sampled from the student policy $\pi_\theta$.
>
> - Accordingly, we have reformulated our theoretical motivation to derive the objective function directly under the training policy $\pi_\theta$.
>
>     - Similar to our original theoretical motivation, we define the *oracle policy*, $\pi_{\text{oracle}}$, as the model's generative process when conditioned on the ground-truth answer $Y_{\text{gt}}$:
>     $$\pi_{\text{oracle}}(h_k) \triangleq \pi_\theta(h_k \mid q, Y_{\text{gt}}, H_{<k})$$
>
>     - Here, instead of relying on the oracle policy assumption, we frame $\pi_{\text{oracle}}$ as representing a "better" policy than the training policy $\pi_{\theta}$ (which generates steps without access to the answer). This perspective aligns with recent works [1,2], which uses ground-truth conditioned policy to generate reasoning steps for training.
>
>     - We demonstrate that maximizing the expected confidence gain $J(\theta)$ is mathematically equivalent to minimizing the KL divergence between the current policy and the superior oracle policy (due to Bayes Rule. Details are in `Appendix G Eq.14`):
>
>        $$J(\theta) = E_{h_k \sim \pi_{\theta}} [C_k] = E_{h_k \sim \pi_{\theta}} \left[ \log \frac{\pi_{\text{oracle}}(h_k)}{\pi_{\theta}(h_k)} \right] = - D_{KL}(\pi_{\theta} \ \| \ \pi_{\text{oracle}}) $$
>
>
> - This derivation provides a concrete theoretical motivation for our proposed method: it proves that by maximizing this reward, the model implicitly imitates the superior reasoning process of the ground-truth conditioned model.
>
> - Additionally, we have moved this theoretical motivation (`Section 4.2`) to `Appendix G`. This allows the main text to focus on the empirical validation and causal analysis of the reward signal, which we agree are the core contributions of the paper.
>
> ---
>
> ### Reference
>
> [1] Zelikman  et al., "Star: Self-taught reasoner bootstrapping reasoning with reasoning." 2024.
>
> [2] Wang et al., "Think or Not? Selective Reasoning via Reinforcement Learning for Vision-Language Models." 2025.

---

> ### Author Response · Authors · 2025-11-28
> **Response to Reviewer 8r9w (3/3)**
>
> > **[Q3]** Because the reward comes from the model’s own probability of the final answer, the model may learn to raise that probability early by adding hints or formatting patterns that correlate with the answer tokens, without improving reasoning quality. PRM works try to avoid such reward hacking with different credit assignment, for example, min form credit or verifier designs that measure true progress. A safety check or a negative control would be useful.
>
> - We appreciate the reviewer for raising this critical point regarding reward hacking. We agree that in Process Reward Model (PRM) training, models often learn to exploit flaws in the fixed reward function.
>
> - We respectfully argue that PACR is **more robust** to hacking than standard PRMs.
>     - PRMs rely on a **fixed external model** (frozen judge). Once the policy identifies a blind spot in the judge (e.g., "always prefers long answers"), it can exploit it indefinitely.
>     - PACR relies on the **evolving policy's own intrinsic belief**.
>         - As detailed in `Section 3`, we measure confidence at each step by appending a fixed answer prefix (e.g., `So the final answer is \boxed{`).
>         - To increase the reward, the model must genuinely increase its *own* probability assigned to the true answer token. It is difficult for a model to "trick itself" into this low-entropy state without actually reducing uncertainty via valid reasoning. **Considering the autoregressive nature of LLMs, if the model merely adds irrelevant hints or formatting, the conditional probability of the answer token is unlikely to sharpen.**
>
> - Finally, we emphasize that PACR is never optimized in isolation. It is strictly coupled with the standard **Outcome-Based Reward**.
>     - If a model learns a hacking strategy (e.g., outputting the answer token early) that degrades reasoning quality, it will likely fail to match the ground truth at the end of the generation (or fail the strict format check).

---

### Official Review · Reviewer_PwTw · 2025-11-03

**Soundness:** 3
**Presentation:** 3
**Contribution:** 2
**Rating:** 4
**Confidence:** 4

**Summary:**

Arguably, dense process rewards can lead to greater improvement with RL; however, acquiring quality process rewards can be costly. This paper introduces PACR, a novel method to enhance Reinforcement Learning with Verifiable Rewards (RLVR) for LLM reasoning. It provides a dense, model-intrinsic reward based on the idea that along a successful reasoning trajectory, the model's confidence in the correct final answer should progressively increase. The correlation between this increase in confidence and outcome correctness implies that it can serve as a process reward.

**Strengths:**

The paper addresses a key problem in RL training: that dense rewards are hard and expensive to acquire.

The paper's presentation is clear. The supporting evidence (observations 1, 2, and 3) for the method is mostly relevant and well thought-out.

The gain from the methods is good over the baseline Dr.DRPO , though in some certain test datasets it is negative.

**Weaknesses:**

The experiment is slightly limited, with only one training dataset and three models, and one baseline algorithm (Dr.GRPO). I would like to see one reasoning model (e.g., DeepSeek-R1-Distill-Qwen-1.5B) tested to see if the effectiveness of your proposed process reward still holds. Also, report the accuracy on AIME 2025.

I dislike the inclusion of Section 4.2, as it makes the proposed method look deep, whereas the key to the proof is really the artificial "oracle policy assumption". I think that Section 4.2 doesn't meaningfully justify your proposed process reward, or you can leave it in the appendix.

The authors only evaluate with pass@1 greedy decoding, yet it is debated whether RL post-trained models increase sampling efficiency (better pass@1) while reducing coverage (lower pass@k for large k compared to the base model). Therefore, the authors should also report $\text{pass@K}$ with a positive temperature.

The author can use this new process-reward for beam search. In my opinion, this is an important experiment, as if beam search leads to improvement, then I'm more confident that RL's improvement comes mainly from the process-reward and not your other design choices (e.g. mixing with outcome reward, mix-max scaling).

**Questions:**

How do the authors justify the process-wise group normalization (in lines 363, before the Min-Max scaling explanation)? Does there exist similar method in the literature? Does the way they divide the sequence into steps of varied lengths have any impact on this?

It is hard for me to read the small colored numbers in Table 1.

What is the setting of Figure 8?

Qualitatively, it would be nice to attempt to find failure cases when high growth doesn't correspond to an informative step, or when low growth doesn't correspond to an uninformative or wrong step. I can see in Figure 12 and 13 steps that have negative, large magnitude growth that are nevertheless informative.

I'm also concerned about the way you divide the sequence into steps. It is very hard to do that nicely by just pattern. I can see that you have many short steps. Did you consider other alternatives?

---

> ### Author Response · Authors · 2025-11-28
> **Response to Reviewer PwTw (1/8)**
>
> We sincerely appreciate your time and constructive comments, which improve our paper. We address your concerns as follows.
>
> ---
> > **[Weakness 1]** The experiment is slightly limited, with only one training dataset and three models, and one baseline algorithm (Dr.GRPO). I would like to see one reasoning model (e.g., DeepSeek-R1-Distill-Qwen-1.5B) tested to see if the effectiveness of your proposed process reward still holds. Also, report the accuracy on AIME 2025.
>
> - We have expanded our experiments in the revised manuscript as follows.
>
> - **DeepSeek-R1-Distill-Qwen-1.5B:** Following the reviewer's suggestion, we integrated **DeepSeek-R1-Distill-Qwen-1.5B** into our experimental setup. We evaluated whether PACR provides additive value to a model that has already undergone extensive reasoning distillation. As detailed in the updated `Table 1`, PACR achieves consistent performance gains even on this strong reasoning baseline. Specifically, while the base model averages **25.4%** and Dr.GRPO improves this to **36.9%**, both **Sparse-PACR (38.9%)** and **Dense-PACR (40.0%)** yield further gains.
>
> - **AIME 2025 Benchmark:** We also incorporated the **AIME 2025** benchmark. We report these results in the revised `Table 1`, showing that our proposed PACR maintains its robustness on this test set.
>
> - **Deepened Evaluation Metrics (Pass@k Analysis):** To ensure our results reflect genuine reasoning improvements rather than just mode collapse, we expanded our evaluation metrics. As detailed in the new `Section 7.2` (`Table 2`), we analyzed **Pass@1 (n=16)** and **Pass@16** with temperature set to $T=1.0$. We found that PACR improves both, confirming that PACR demonstrates a genuine expansion of reasoning capability.
>
>
> - **Comparison with Dynamic Sampling (DAPO):** To address the concern regarding baseline diversity, we expanded our comparison to include a stability-oriented baseline. Specifically, we integrated the **dynamic sampling strategy** from DAPO, which is explicitly designed to solve the vanishing advantage problem by maintaining a consistent effective batch size. As detailed in the added `Section 7.3 (Table 3)`, this comparison allows us to disentangle the source of our gains.
> 	- We observe that while dynamic sampling improves the Dr.GRPO baseline (raising average accuracy from **36.8%** to **39.0%**), **Dense-PACR (40.0%) consistently outperforms it**. Furthermore, combining both methods yields the highest performance (**40.6%**). This confirms that the performance boost stems from the **informative quality of the dense signal** ($C_k$) itself, providing directional guidance, rather than solely from improved training stability.
>
> - **Generalization to Logical Reasoning:** Finally, to demonstrate that our method's effectiveness extends beyond mathematics, we conducted additional experiments on **Logical Reasoning** tasks. As detailed in `Section 7.7 (Table 4)`, we observe a critical failure mode for the standard outcome-based baseline (Dr.GRPO), which **actually degrades performance** compared to the untrained base model on logical benchmarks. In contrast, PACR **recovers and exceeds** the base model's performance.

---

> ### Author Response · Authors · 2025-11-28
> **Response to Reviewer PwTw (2/8)**
>
> > **[Weakness 2]** I dislike the inclusion of Section 4.2, as it makes the proposed method look deep, whereas the key to the proof is really the artificial "oracle policy assumption". I think that Section 4.2 doesn't meaningfully justify your proposed process reward, or you can leave it in the appendix.
>
> - In response to your suggestion, we have moved the formal proof (`Section 4.2`) to `Appendix G`. This allows the main text to focus on the empirical validation and causal analysis of the reward signal, which we agree are the core contributions of the paper.
>
> - We agree that our original formulation, which analyzed properties under the oracle policy, did not fully address the dynamics of the actual training process where steps are sampled from the student policy $\pi_\theta$.
>
> - Accordingly, we have reformulated our theoretical motivation to derive the objective function directly under the training policy $\pi_\theta$.
>
>     - Similar to our original theoretical motivation, we define the *oracle policy*, $\pi_{\text{oracle}}$, as the model's generative process when conditioned on the ground-truth answer $Y_{\text{gt}}$:
>     $$\pi_{\text{oracle}}(h_k) \triangleq \pi_\theta(h_k \mid q, Y_{\text{gt}}, H_{<k})$$
>
>     - Here, instead of relying on the oracle policy assumption, we frame $\pi_{\text{oracle}}$ as representing a "better" policy than the training policy $\pi_{\theta}$ (which generates steps without access to the answer). This perspective aligns with recent works [1,2], which uses ground-truth conditioned policy to generate reasoning steps for training.
>
>     - We demonstrate that maximizing the expected confidence gain $J(\theta)$ is mathematically equivalent to minimizing the KL divergence between the current policy and the superior oracle policy (due to Bayes Rule. Details are in `Appendix G Eq.14`):
>
>         $$J(\theta) = E_{h_k \sim \pi_{\theta}} [C_k] = E_{h_k \sim \pi_{\theta}} \left[ \log \frac{\pi_{\text{oracle}}(h_k)}{\pi_{\theta}(h_k)} \right] = - D_{KL}(\pi_{\theta} \ \| \ \pi_{\text{oracle}}) $$
>
> - This derivation provides a concrete theoretical motivation for our proposed method: it proves that by maximizing this reward, the model implicitly imitates the superior reasoning process of the ground-truth conditioned model.
>
> ---
>
> ### Reference
>
> [1] Zelikman  et al., "Star: Self-taught reasoner bootstrapping reasoning with reasoning." 2024.
>
> [2] Wang et al., "Think or Not? Selective Reasoning via Reinforcement Learning for Vision-Language Models." 2025.

---

> ### Author Response · Authors · 2025-11-28
> **Response to Reviewer PwTw (3/8)**
>
> > **[Weakness 3]** The authors only evaluate with pass@1 greedy decoding, yet it is debated whether RL post-trained models increase sampling efficiency (better pass@1) while reducing coverage (lower pass@k for large k compared to the base model). Therefore, the authors should also report with a positive temperature.
>
> - To address this, we conducted an analysis using the **Qwen3-4B** model with a positive sampling temperature ($T=1.0$). We evaluated performance using two metrics to disentangle the sources of improvement:
>     - **Pass@1 (n=16):** Calculated as the average accuracy across 16 sampled trajectories. This proxies \textit{sampling efficiency} (sharpening probability on correct paths).
>     - **Pass@16:** Calculated as the probability that at least one of the 16 samples is correct. This proxies \textit{intrinsic reasoning capability} (expanding the manifold of solvable problems).
>
> - As added in `Section 7.2 (Table 2)`, Dense-PACR achieves consistent gains across both metrics.
>     - **Efficiency:** While the Dr.GRPO baseline achieves **46.4%** Pass@1 (n=16), Dense-PACR improves this to **50.6%** (a **+4.2%** gain), confirming improved sampling efficiency.
>     - **Capability:** Crucially, for Pass@16, Dense-PACR improves from the baseline's **64.3%** to **67.1%** (a **+2.8%** gain).

---

> ### Author Response · Authors · 2025-11-28
> **Response to Reviewer PwTw (4/8)**
>
> > **[Weakness 4]** The author can use this new process-reward for beam search. In my opinion, this is an important experiment, as if beam search leads to improvement, then I'm more confident that RL's improvement comes mainly from the process-reward and not your other design choices (e.g. mixing with outcome reward, mix-max scaling).
>
> - We are grateful for this excellent suggestion. We agree that using the reward signal for test-time search is the most rigorous way to disentangle the intrinsic quality of the reward from the dynamics of RL training.
>
> - **New Experiment (Beam Search with PACR):** To address this, we have integrated this experiment into the revised `Section 4.1` (see Paragraph b: *`Causal Utility: Does guiding generation with this bias improve accuracy during inference?`*).
>     - Specifically, we implemented a **step-wise Best-of-N search strategy** (equivalent to beam search with width 1). At each reasoning step, we sampled $N=8$ candidates and selected the path maximizing the raw confidence gain ($C_k$).
>     - Crucially, this setup uses the **pure confidence gain as a process reward signal** without the outcome mixing or min-max normalization used during RL training, directly testing the reviewer's hypothesis.
>
> - As shown in the added `Figure 2-(b)` of `Section 4.1`, the PACR-guided search during inference significantly outperforms standard sampling baselines across multiple temperatures ($T=0.8, 0.9, 1.0$).
>
> - As the reviwer pointed out, this result suggests that the performance gains are driven by the **informative nature of the process reward itself**, rather than being an artifact of specific RL implementation details.

---

> ### Author Response · Authors · 2025-11-28
> **Response to Reviewer PwTw (5/8)**
>
> > **[Q1]** How do the authors justify the process-wise group normalization (in lines 363, before the Min-Max scaling explanation)? Does there exist similar method in the literature? Does the way they divide the sequence into steps of varied lengths have any impact on this?
>
> - We thank the reviewer for this insightful question. Motivated by the principles of time-dependent baseline calculation [1], we utilized process-wise normalization to address temporal distribution shifts in reward distributions. Standard trajectory-level normalization can be problematic in such contexts, as it may dilute or distort the significance of specific time steps when rewards vary significantly over time.
> - In our specific context where a trajectory consisting of sound reasoning steps exhibits monotonic confidence growth, the calculated returns tend to be systematically larger in the early stages of the episode compared to the later stages.
> - Consequently, applying trajectory-level normalization would bias the advantage estimation, assigning disproportionately values to earlier reasoning steps simply due to their temporal position.
> - By employing process-wise normalization, we ensured that each problem-solving step is evaluated based on its relative quality within its specific temporal context.
> - We acknowledge the reviewer's valid point that varied sequence lengths can increase the variance of normalization statistics, potentially introducing noise. However, we determined that the negative impact of inaccurate evaluations caused by trajectory-level baseline would be more detrimental to learning performance than the noise introduced by the normalization process itself.
>
> ---
>
> ### Reference
>
> [1] Peters et al., "Reinforcement learning of motor skills with policy gradients." 2008.

---

> ### Author Response · Authors · 2025-11-28
> **Response to Reviewer PwTw (6/8)**
>
> > **[Q2]** It is hard for me to read the small colored numbers in Table 1.
>
> - To address this, we have revised `Table 1` by applying bold formatting to the colored improvement numbers to enhance visibility. Additionally, we have simplified the table layout by displaying these improvement metrics only for the Average column, reducing visual clutter and ensuring the table is less condensed.
>
> > **[Q3]** What is the setting of Figure 8?
>
> - The training curves in `Figure 8` (which is now `Figure 7` in the revised manuscript) were generated using Qwen2.5-Math-1.5B. We have updated the figure caption in the revised manuscript to explicitly state this experimental setting.

---

> ### Author Response · Authors · 2025-11-28
> **Response to Reviewer PwTw (7/8)**
>
> > **[Q4]** Qualitatively, it would be nice to attempt to find failure cases when high growth doesn't correspond to an informative step, or when low growth doesn't correspond to an uninformative or wrong step. I can see in Figure 12 and 13 steps that have negative, large magnitude growth that are nevertheless informative.
>
> - We thank the reviewer for their careful examination of the qualitative examples. We agree that characterizing these mismatches between signal and quality is crucial for understanding the method's behavior.
>
> - **Analysis of Near Zero Growth for Valid Steps (Confidence Saturation):** In response to this comment, we explicitly analyzed cases where valid, necessary steps receive near-zero or slightly negative gains. We term this phenomenon **Confidence Saturation** and have added a detailed discussion in `Appendix E`.
>     - We observe that after a pivotal step yields a massive confidence gain (e.g., $C_k > +4.0$), the immediately subsequent steps often exhibit near-zero gains ($C_{k+1} \approx 0$), even when they represent valid algebraic manipulations.
>     - We argue that this correctly reflects the **information dynamics** of reasoning. Once the pivotal insight is established, the *remaining uncertainty* regarding the final answer drops to near zero. Therefore, subsequent execution steps, while necessary, provide minimal *new* information about the ground truth.
>
> - Given the reviewer's observation that these steps are nevertheless informative, it is critical that our min-max normalization strategy (as discussed in `Section 7.5: Anlaysis on Advantage Formulation (Impact of Penalizing Intermediate Steps)`) does not actively penalize them.
>     - By incorporating a min-max normalization, we normalize the reward in the $[0, 1]$ range.
>     - By normalizing rewards to the $[0, 1]$ range, these saturated steps (where $C_k \approx 0$) are mapped to a **neutral reward** rather than a negative penalty.

---

> ### Author Response · Authors · 2025-11-28
> **Response to Reviewer PwTw (8/8)**
>
> > **[Q5]** I'm also concerned about the way you divide the sequence into steps. It is very hard to do that nicely by just pattern. I can see that you have many short steps. Did you consider other alternatives?
>
> - We thank the reviewer for raising this important design question. We agree that defining the fundamental unit of reasoning is non-trivial. In response, we have added a detailed discussion in `Appendix B` comparing our approach to alternative segmentation paradigms.
>
> - **Why we chose Heuristic Segmentation:** We explicitly considered two main alternatives but prioritized the delimiter-based heuristic for specific reasons:
>     - **vs. SFT-Constrained Segmentation:** Methods that train models to output explicit tags (e.g., `<step>`) require high-quality human data. Since our goal is to follow the **DeepSeek-R1-Zero** [1] paradigm, incentivizing reasoning directly from the base model without relying on supervised cold-start data, this approach was incompatible.
>     - **vs. Entropy-Based Segmentation:**  While methods that slice steps based on intrinsic signals (e.g., perplexity spikes) [2] are theoretically promising, they remain an **emerging research direction** rather than an established standard.
>     - Therefore, we adopted the **standard delimiter-based heuristic** used in most prior process reward literatures [3,4,5].
>
> - **Chunk-Level PACR (Efficiency Trade-off):** Moreover, to potentially mitigate the overhead in long-context scenarios, we investigated a **Chunk-Level PACR** strategy, where rewards are computed only at coarse intervals (every $k$ steps).
>     - As detailed in `Table 7` (`Appendix I`), aggregating every two steps ($k=2$) reduces the reward computation frequency by **50%** while achieving an accuracy of **39.7%**, effectively on-par with the dense baseline (Dense-PACR, 40.0%).
>     - Even with a coarser granularity of $k=4$ (reducing overhead by **75%**), the model achieves **38.7%**, which remains significantly higher than the Dr.GRPO baseline (36.8%). This confirms that PACR offers a flexible trade-off: users can tune the chunk size to fit their computational budget without sacrificing the method's superiority over sparse outcome-based reward.
> ---
>
> ### Reference
>
> [1] Guo et al., "Deepseek-r1: Incentivizing reasoning capability in llms via reinforcement learning." 2025.
>
> [2] Liu et al., "Adaptivestep: Automatically dividing reasoning step through model confidence." 2025.
>
> [3] Yang et al., "Beyond the First Error: Process Reward Models for Reflective Mathematical Reasoning." 2025,
>
> [4] Zengh et al., "Versaprm: Multi-domain process reward model via synthetic reasoning data." 2025
>
> [5] Lee et al., "Rethinking Reward Models for Multi-Domain Test-Time Scaling." 2025

---

### Official Review · Reviewer_5e8n · 2025-11-03

**Soundness:** 2
**Presentation:** 3
**Contribution:** 2
**Rating:** 4
**Confidence:** 4

**Summary:**

The paper introduces PACR, a dense, model-intrinsic reward for reinforcement learning in LLM reasoning. Instead of relying solely on sparse, outcome-based signals, PACR rewards stepwise increases in the model’s confidence in the ground-truth answer. The authors show that this “confidence ascent” acts as a strong training signal for training and can empirically enhance RL training performance. Experiments on multiple math-reasoning benchmarks (e.g., MATH500, AIME24, AMC23) demonstrate that PACR improves RL training performance.

**Strengths:**

1. The problem addressed in this paper is important. Incorporating more dense reward information into current RL pipelines remains an underexplored direction.


2. The proposed idea is reasonable and aligns with recent studies showing that a model’s reasoning confidence often correlates with the correctness of its answers, suggesting that such confidence signals could be valuable for training.


3. The writing is clear and easy to follow.

**Weaknesses:**

1. The approach for determining the reasoning step appears rather ad-hoc. It is unclear how this mechanism would transfer to other domains such as code generation, where the output often contains many new lines. Would this lead to an excessive number of reasoning steps for tasks involving long-context generation?

2. The training process seems to introduce additional computational overhead, particularly as the generation length increases, which could significantly inflate the number of reasoning steps. A detailed analysis of the training cost and the number of reasoning steps would help clarify the efficiency of the proposed method.

3. The experiments use a maximum generation length of only 3k tokens, which is much shorter than the 40k-token context supported by the Qwen3-4B model. Given that the proposed method may introduce substantial overhead for long-context scenarios, it would be valuable to examine its scalability with respect to context length to better assess its effectiveness.

4. Another possible explanation for the observed performance improvement could be more stable training due to the absence of zero-advantage samples. In vanilla GRPO, many responses have zero advantage and thus contribute nothing to training (though dynamic sampling can alleviate this issue). Since PACR avoids this, it would be helpful to compare against (or combine with) GRPO with dynamic sampling to determine whether the gains primarily stem from denser rewards or from improved training stability.

**Questions:**

listed in weakness

---

> ### Author Response · Authors · 2025-11-28
> **Response to Reviewer 5e8n (1/4)**
>
> We sincerely appreciate your time and constructive comments, which improve our paper. We address your concerns as follows.
>
> ---
> > **[Q1]** The approach for determining the reasoning step appears rather ad-hoc. It is unclear how this mechanism would transfer to other domains such as code generation, where the output often contains many new lines. Would this lead to an excessive number of reasoning steps for tasks involving long-context generation?
>
> - We appreciate the reviewer for raising this insightful question regarding the definition of a "reasoning step" and its generalization.
>
> - Regarding the method of determining a step, we explicitly adopt the segmentation heuristics used in recent process reward literature [1,2,3], where they utilize this delimiter-based approach (e.g., newlines/periods) to define reasoning steps. To provide comprehensive context, we have added a discussion in `Appendix B` that surveys alternative segmentation strategies in the literature.
>
> - **Clarification on Code Domains:** Regarding the transfer to code, we find it important to distinguish between **pure code generation** and **code-aided reasoning** (using code interpreters to solve problems). Given our paper's focus on reasoning capabilities, the most relevant extension is code-aided reasoning, where the model generates code blocks to perform calculations or verify logic [4,5,6].
>
>
> - **Adaptability to Code-Aided Reasoning:** In this context, we agree that a strict newline rule would be suboptimal. Instead, the *logical unit* of reasoning becomes the entire code execution block rather than individual lines of code. We would define a "step" as the entire content within the code delimiters (e.g., ```import numpy ... ```).Executing this block and obtaining an output constitutes a single, discrete event of uncertainty reduction, making the full block the appropriate granularity for the $C_k$ calculation.
>
> - **Future Work and Generalization:** While the "Code Block-as-a-Step" approach is fully compatible with our PACR framework, our current experiments prioritize Natural Language Chain-of-Thought (CoT) reasoning. We therefore leave the empirical validation of PACR for code domains to future work. We have added this dicussion regarding code-aided reasoning in `Appendix A.1: Limitations and Future Work`.
> - **However, to demonstrate the method's applicability beyond mathematical reasoning, we conducted additional experiments on the Logical Reasoning domain.** We provide these experimental results in `Section 7.7` (`Table 4`), confirming that the benefits of our proposed dense confidence reward extend to broader reasoning domains.
>
> ---
>
> ### Reference
>
> [1] Yang et al., "Beyond the First Error: Process Reward Models for Reflective Mathematical Reasoning." 2025,
>
> [2] Zengh et al., "Versaprm: Multi-domain process reward model via synthetic reasoning data." 2025
>
> [3] Lee et al., "Rethinking Reward Models for Multi-Domain Test-Time Scaling." 2025
>
> [4] Gao et al., "Pal: Program-aided language models." 2023
>
> [5] Yu et al., "Siam: Self-improving code-assisted mathematical reasoning of large language models." 2024
>
> [6] Yang et al., "Code to think, think to code: A survey on code-enhanced reasoning and reasoning-driven code intelligence in llms." 2025

---

> ### Author Response · Authors · 2025-11-28
> **Response to Reviewer 5e8n (2/4)**
>
> > **[Q2]** The training process seems to introduce additional computational overhead, particularly as the generation length increases, which could significantly inflate the number of reasoning steps. A detailed analysis of the training cost and the number of reasoning steps would help clarify the efficiency of the proposed method.
>
> - We appreciate the reviewer for raising this practical concern regarding computational overhead. We acknowledge that while the confidence gain ($C_k$) are computed via batched forward passes, calculating them at every reasoning step intrinsically incurs additional inference cost during the rollout phase that scales linearly with the number of steps.
>
> - **Accuracy vs. Wall-Clock Efficiency:** However, while the *per-step* compute is higher for PACR due to the confidence gain calculation, the *sample efficiency* is superior to the baseline.
>     - We have added a detailed efficiency analysis in `Figure 8` (`Section 7.6`). As shown in `Figure 8-(a)`, PACR indeed has a higher wall-clock time per training step.
>     - However, `Figure 8-(b)` demonstrates that when accuracy is plotted against **total wall-clock training time**, the PACR curve lies above the baseline. This indicates that the acceleration in learning provided by the confidence gain signal effectively outweighs the overhead of the additional forward passes.
>
> - **Chunk-Level PACR (Efficiency Trade-off):** Moreover, to potentially mitigate the overhead in long-context scenarios, we investigated a **Chunk-Level PACR** strategy, where rewards are computed only at coarse intervals (every $k$ steps).
>     - As detailed in `Table 7` (`Appendix I`), aggregating every two steps ($k=2$) reduces the reward computation frequency by **50%** while achieving an accuracy of **39.7%**, effectively on-par with the dense baseline (Dense-PACR, 40.0%).
>     - Even with a coarser granularity of $k=4$ (reducing overhead by **75%**), the model achieves **38.7%**, which remains significantly higher than the Dr.GRPO baseline (36.8%). This confirms that PACR offers a flexible trade-off: users can tune the chunk size to fit their computational budget without sacrificing the method's superiority over sparse outcome-based reward.

---

> ### Author Response · Authors · 2025-11-28
> **Response to Reviewer 5e8n (3/4)**
>
> > **[Q3]** The experiments use a maximum generation length of only 3k tokens, which is much shorter than the 40k-token context supported by the Qwen3-4B model. Given that the proposed method may introduce substantial overhead for long-context scenarios, it would be valuable to examine its scalability with respect to context length to better assess its effectiveness.
>
> - We thank the reviewer for highlighting the importance of validating our method in long-context reasoning scenarios. To address this, we conducted an additional experiment using **R1-Distill-Qwen-1.5B**, increasing the maximum generation length to **8,192 tokens**.
>
> - As reported in the updated `Table 1`, PACR maintains its robust performance advantage even in this long-context regime. The average accuracies across benchmarks are:
>     - **R1-Distill-Qwen-1.5B:** 25.4%
>     - **Dr.GRPO (Baseline):** 36.9%
>     - **Sparse-PACR:** 38.9%
>     - **Dense-PACR:** 40.0%
>
> - Dense-PACR achieves a **+3.1%** absolute improvement over the Dr.GRPO baseline. This confirms that the effectiveness of the dense confidence signal scales well with context length, likely because dense supervision helps maintain reasoning coherence over long horizons where sparse rewards struggle with credit assignment.

---

> ### Author Response · Authors · 2025-11-28
> **Add: Response to Reviewer 5e8n (4/4)**
>
> > **[Q4]** Another possible explanation for the observed performance improvement could be more stable training due to the absence of zero-advantage samples. In vanilla GRPO, many responses have zero advantage and thus contribute nothing to training (though dynamic sampling can alleviate this issue). Since PACR avoids this, it would be helpful to compare against (or combine with) GRPO with dynamic sampling to determine whether the gains primarily stem from denser rewards or from improved training stability.
>
>
> - To strictly verify whether our gains stem primarily from non-zero advantage (stability) or better guidance (dense rewards), we added the **dynamic sampling** strategy from DAPO [1] as a rigorous baseline. This method resolves the vanishing advantage by resampling trajectories until the group contains diverse outcomes.
>
> - As detailed in `Section 7.3 (Table 3)` we applied this comparison using the **R1-Distill-Qwen-1.5B** model with long-context generation (8k). We observe that while dynamic sampling indeed boosts the Dr.GRPO baseline (raising accuracy from 36.8\% to 39.0\%), **Dense-PACR (40.0\%) outperforms this stabilized baseline**. Furthermore, combining both methods yields the highest performance (40.6\%).
>
> - The fact that PACR outperforms the stable-gradient baseline confirms that the **confidence growth signal ($C_k$)** provides necessary directional guidance that provides benefits beyond simply mitigating the vanishing advantage problem.
>
> ---
>
> ### Reference
>
> [1] Yu et al., "DAPO: An Open-Source LLM Reinforcement Learning System at Scale." 2025.

---

### Author Response · Authors · 2025-12-03
**General Response (2/2)**

**V. Computational Efficiency Analysis**

- **`Section 7.6, Figure 8` (Computation Overhead)**: We provide analysis on the computation overhead. While our proposed PACR incurs a higher per-step cost, our **Time-to-Convergence** analysis demonstrates that the superior sample efficiency effectively outweighs this overhead, resulting in a more favorable accuracy-per-wall-clock trade-off.

- **`Appendix I, Table 7` (Chunk-Level PACR)**: To further mitigate computational overhead, we introduced a Chunk-Level PACR strategy (Step Coarsening). We demonstrated that grouping every 2 steps ($k=2$) reduces the reward computation frequency by 50% while maintaining performance (39.7%) effectively on-par with the dense baseline (Dense-PACR, 40.0%). This suggests that PACR retains its efficacy even with coarser step granularity, allowing users to tune the reward frequency to fit computational budgets without significantly degrading performance.

**VI. Reasoning Segmentation Strategy**

To clarify our segmentation method and justify our design choice, we added the following discussion.
- **`Appendix B` (Segmentation Strategy)**: We added a survey of reasoning decomposition methods. We contextualize our choice of the heuristic delimiter-based strategy against SFT-constrained and entropy-based alternatives, justifying it based on standardization across prior literature and compatibility with the unsupervised R1-Zero paradigm.

**VII. Failure Case Discussion**

To provide deeper insight into metric behavior, including analysis of potential failure mode, we added the following discussion.
- **`Appendix E` (Failure Case Analysis)**: We added a dicussion of the **Confidence Saturation** phenomenon, where valid execution steps following a pivotal insight exhibit near-zero confidence gains. We discuss why this reflects accurate information dynamics and explain how our **Min-Max normalization** strategy mitigates this by ensuring such steps are not penalized.

---

### Author Response · Authors · 2025-12-03
**General Response (1/2)**

We sincerely thank the **Area Chair** and **all reviewers** for their time, and insightful feedback. These reviews have been instrumental in refining both the empirical rigor and the theoretical clarity of our work.

We believe we have comprehensively addressed all the concerns raised. In the revised manuscript, we have incorporated these revisions (highlighted in `blue`), which are organized by category below:

---

**I. Rigorous Validation of Observations and GPT Annotation Reliability**

To address concerns regarding circularity and subjectivity in **`Section 4: Is Ground-Truth Confidence Growth a Useful Inductive Bias?`**, we replaced descriptive observations with causal experiments and validated our labeling protocol used in the analysis.

- **`Section 4`** & **`Figure 2` (Signal Validation)**: We have restructured this section to provide rigorous empirical validation of the reward signal. We replaced the original observations with two new quantitative experiments:
    - `(a) Granular Step-Quality Analysis`, which confirms that confidence gains correlate with local reasoning quality (Good vs. Bad steps)
    - `(b) Test-Time Causal Utility`, which demonstrates that actively guiding beam search with PACR significantly improves inference accuracy validating it as a process reward.

- **`Appendix F` (Label Acquisition)**: We provided details on the annotation protocol used for the step quality analysis in `Section 4.1`. This includes the exact prompt design and an inter-annotator agreement study for validating the GPT-labeled quality of steps.

**II. Theoretical Refinement**

- **`Appendix G` (Theoretical Motivation)**: We relocated the formal mathematical derivation (originally Section 4.2) to `Appendix G` and refined the main text to focus on the conceptual framework of Uncertainty Reduction. We also revised the theoretical motivation to enhance rigor, explicitly **proving that maximizing the PACR reward is equivalent to minimizing the *KL divergence* between the student policy and the ground-truth conditioned "Oracle Policy."**


**III. Expanded Experimental Scope & Generalization**

To demonstrate robustness, we expanded our evaluation across more models, more benchmarks, and broader domains.

- **`Section 7.1, Table 1` (Additional Model and Benchmark)**: We expanded our main experimental results to include **R1-Distill-Qwen-1.5B** and the recent **AIME 2025** benchmark.

- **`Section 7.3, Table 3` (DAPO)**: We introduced a rigorous comparison with the **dynamic sampling strategy** (adopted from **DAPO**). This expansion allows us to disentangle the benefits of **dense supervision** from **gradient stability**, confirming that PACR outperforms baselines that rely solely on mitigating the vanishing advantage problem.

- **`Section 7.7, Table 4` (Logical Reasoning)**: We expanded our experimental setup to logical reasoning domain, showing PACR generalizes beyond math-heavy tasks.

- **`Figure 7` (Training Dynamics)**: We updated the caption to explicitly specify the experimental setting (Qwen2.5-Math-1.5B).

- **`Appendix A.1` (Future Extensions)**: We included a discussion on **Code-Aided Reasoning**, demonstrating how the framework could extend to executable domains.

**IV. Robustness and Statistical Reliability**

To show that our performance gains reflect genuine reasoning improvements rather than sampling artifacts or stochastic variance, we included evaluation results using pass@k and standard deviation.

- **`Section 7.2, Table 2` (Pass@k Analysis)**: We introduced a detailed analysis of **Pass@k** metrics at sampling temperature $T=1.0$. These results disentangle **sampling efficiency** from **intrinsic capability**, suggesting that PACR enhances the model's fundamental reasoning power without inducing mode collapse.

- **`Appendix H` (Standard Deviation)**: We provided the standard deviation analysis corresponding to the main results in `Table 1`.

---

### Meta-Review · Area_Chair_YX4h · 2026-01-13

**Summary:**

The reviewers broadly agreed that the paper introduces a clear and intuitively appealing idea—using stepwise increases in model confidence in the ground-truth answer as a dense, intrinsic reward for RL-based reasoning. Strengths consistently noted include conceptual simplicity, avoidance of external reward models, and empirical gains across several math reasoning benchmarks.

However, the main concerns motivating caution were:

- Theoretical justification: Multiple reviewers (especially Reviewer 1VLi and 8r9w) criticized the original theoretical argument as relying too heavily on an oracle-policy assumption, making it appear tautological and weakly connected to actual training dynamics.

- Empirical scope and robustness: Reviewers questioned whether the gains were sufficiently large, stable, and generalizable—particularly beyond math reasoning, across more models, longer contexts, and alternative evaluation metrics (e.g., pass@k).

- Alternative explanations for gains: Some reviewers raised the possibility that improvements stem from improved training stability (non-zero advantages) rather than the informational quality of the confidence-based reward itself.

- Practical concerns: These included computational overhead, step segmentation heuristics, scalability to long-context or code-like outputs, and potential reward hacking (models inflating confidence without better reasoning).

**Reviewer Concerns:**

Concerns Largely Addressed by the Rebuttal

- Theoretical clarity:
The authors moved the contentious oracle-based proof to the appendix and reframed the theory as KL minimization toward a ground-truth-conditioned policy, addressing complaints about overclaiming and presentation.

- Empirical breadth and robustness:
Added experiments on:

An additional strong reasoning model (R1-Distill-Qwen-1.5B),

A new benchmark (AIME 2025),

Logical reasoning tasks beyond math,

Pass@k metrics at non-zero temperature, and

Long-context settings (up to 8k tokens).

- Alternative explanations (training stability):
Direct comparison with dynamic sampling (DAPO) showed PACR outperforming stability-focused baselines and achieving best results when combined, strengthening the claim that dense confidence guidance adds value.

- Causality of the reward signal:
New test-time beam-search-style experiments using PACR alone (no RL, no normalization) demonstrated inference-time gains, directly supporting PACR as a meaningful process reward.

- Computational overhead and segmentation:
Detailed wall-clock analysis, chunk-level PACR variants, and expanded discussion of segmentation choices mitigated practical concerns.

- Annotation reliability:
Inter-annotator agreement studies and human–model alignment analyses addressed concerns about GPT-based step-quality labels.

Concerns Partially or Not Fully Resolved

- Strength of empirical gains:
While broader and more consistent, the improvements remain modest (≈2–4% absolute). Some reviewers may still view this as limited given added complexity.

- Generality beyond verifiable-answer tasks:
PACR still fundamentally depends on a known ground-truth answer; extension to open-ended QA or proofs with multiple valid forms remains an acknowledged limitation.

- Reward hacking risk:
The rebuttal provides a plausible argument for robustness, but no explicit negative-control experiment fully rules out subtle forms of confidence manipulation.

- Comparison to trained PRMs or other dense-reward methods:
The authors justify exclusion on data-regime grounds, but some reviewers may still see this as an incomplete empirical positioning.

**Reviewer Scores:**

Reviewer 5e8n:
Likely to increase from 4 → 6 (borderline accept), given that all four major concerns—overhead, long-context scalability, dynamic sampling, and segmentation—were directly and empirically addressed.

Reviewer PwTw:
Likely to increase from 4 → 6, as requested experiments (pass@k, beam search, broader models/benchmarks, theoretical relocation) were carried out thoroughly and align closely with their suggestions.

Reviewer 8r9w:
Possibly maintain 4 Generalization beyond math, theoretical reframing, and discussion of reward hacking address most concerns, though some residual skepticism may remain.

Reviewer 1VLi:
Likely to increase from 2 → 4. While the rebuttal substantially improves theory, causality, and empirical validation, this reviewer may still view the gains as modest and the core idea as insufficiently novel or risky.

---

### Decision · Program_Chairs · 2026-01-26

Reject